# Relatively stable pressure effects and time-increasing thermal contraction control Heber geothermal field deformation

Guoyan Jiang [1] ✉, Andrew J. Barbour [2] ✉, Robert J. Skoumal [3], Kathryn Materna[3], Joshua Taron[3] & Aren Crandall-Bear[4]

Due to geological complexities and observational gaps, it is challenging to identify the governing physical processes of geothermal field deformation including ground subsidence and earthquakes. In the west and east regions of the Heber Geothermal Field (HGF), decade-long subsidence was occurring despite injection of heat-depleted brines, along with transient reversals between uplift and subsidence. These observed phenomena contradict current knowledge that injection leads to surface uplift. Here we show that high-yield production wells at the HGF center siphon fluid from surrounding regions, which can cause subsidence at low-rate injection locations. Moreover, the thermal contraction effect by cooling increases with time and eventually overwhelms the pressure effects of pressure fluctuation and poroelastic responses, which keep relatively stable during geothermal operations. The observed subsidence anomalies result from the siphoning effect and thermal contraction. We further demonstrate that thermal contraction dominates long-term trends of surface displacement and seismicity growth, while pressure effects drive near-instantaneous changes.

Technologies to utilize natural geothermal systems as a source of renewable, carbon-free energy have existed for many decades. However, it is well-documented that geothermal energy production can cause significant geohazards like earthquakes and ground subsidence, which threaten the sustainability of long-term operations and expose nearby populations to potentially damaging impacts[1–4]. Various physical mechanisms have been suggested for the geohazards, including reservoir pressure depletion associated with fluid loss[5,6], poroelastic contraction[7,8], thermoelastic contraction[9–11], and aseismic fault slip[12,13]. Such mechanisms can be active concurrently, but it is challenging to quantify their relative strengths and spatiotemporal evolution features, especially for large-scale fields with complex operation history.

The Heber Geothermal Field (HGF) is a representative field located in the Imperial Valley, southern California, USA, where geothermal resources are abundant (Fig. 1a). Many large energy-producing fields, like the Salton Sea Geothermal Field, have been set up since the early

1980s. All these fields operate in regimes with high crustal strain rates and heat flow owing to the tectonic transition from strike-slip to ridge-transform style tectonics[8,12]. The geohazards associated with geothermal operations in the valley have always been a major concern in both scientific and industrial communities[14,15].

Hydrothermal flow at the HGF is controlled by a matrix permeability structure consisting of a shallow sedimentary reservoir and three faults[16] (Fig. 1b). The upper surface of the reservoir is bound by clay caprock; and two northwest-trending dextral strike-slip faults (plate boundary and subsidiary faults) control its lateral extent. The boundary fault, also known as the Dixieland fault, is part of an active, blind extension of the Superstition Hills fault[17–20]. In addition, there is a northeast-trending normal-slip fault, with a dip angle of ~77° below 1.68 km, which is a primary pathway for deep, high-temperature brine upwelling into the shallow reservoir. We thus refer to it as the feeder fault. With the effect of regional groundwater flow from natural

---

[1]School of Geodesy and Geomatics, Hubei Luojia Laboratory, Wuhan University, Wuhan, China. [2]US Geological Survey, Vancouver, WA, USA. [3]US Geological Survey, Moffett Field, CA, USA. [4]University of Nevada, Reno, NV, USA. ✉e-mail: gyjiang@whu.edu.cn; abarbour@usgs.gov

recharge, the overall shape of the isothermal contour of 182 °C within the reservoir resembles a lopsided mushroom above the depth of ~2 km[16,21].

The HGF resource was initially estimated to support a capacity of up to 500 MW of electrical power generation. One double flash power plant and one binary power plant were completed in 1985; however, due to low production rates, the binary plant designed with a supercritical cycle and a single turbine-generator was discontinued in 1987 and subsequently decommissioned[22]. In 1992, the binary plant was transferred into a subcritical cycle incorporating 6 modules with 33

MWe net output. At present, the field has one double flash and three binary plants, with a capacity of 89 MWe[23]. A total of 69 vertical and deviated wells were drilled during the development history (Fig. 2a), mainly in the years of 1985, 1987, 1993, and 2006–2009 (Supplementary Table 1). Of these, there are 28 injection and 31 extraction wells as well as 10 wells that were first operated for ~2 years of heat extraction, and then used for reinjecting heat-depleted brines since 1993.

Along with the complex operation history, injection, and production fluid temperatures vary with location and time (Fig. 2b). In the west and east regions, where only fluid injection occurs, the average

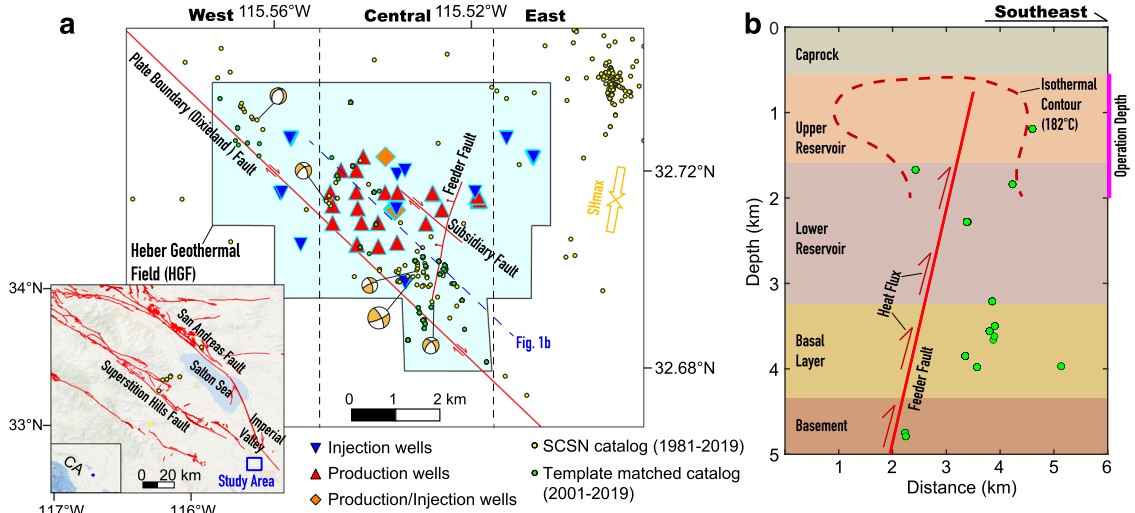

**Fig. 1 | Geologic settings of the Heber Geothermal Field. a** Spatial distribution of operation wells and faults. Yellow and green dots show the events detected by Southern California Seismic Network (SCSN, https://doi.org/10.7914/SN/CI) and with the method of template matching (see Methods for more details). Beachballs show the focal mechanisms of five M > 2 events in the SCSN catalog. The study area is divided into three regions (west, central, and east) to investigate temporal variations of surface displacement. The orientation of the maximum horizontal principal stress ($S_{Hmax}$) is from the SCEC Community Stress Model (https://www.scec.

org/research/csm), which compiles multiple sources of information. **b** Simplified hydro-thermo-geological profile of the HGF. The feeder fault serves as an upwelling channel of heat flux, with the dashed line showing the 182 °C isothermal contour[16]. Circles show earthquake locations projected along the profile in Fig. 1a. The depths of open-hole sections for injection and production wells range from 0.5 to 2 km (pink vertical bar), except for six wells with open-hole depths exceeding 2 km but less than 3.2 km (Supplementary Table 1).

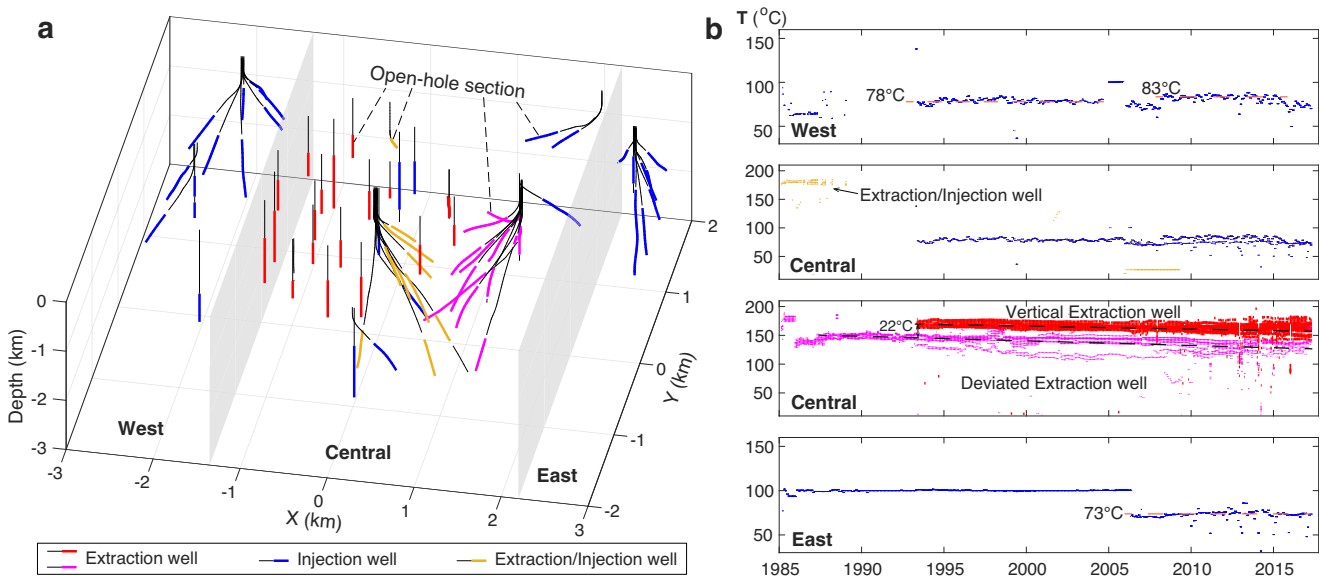

**Fig. 2 | Operation wells and temperature records of the Heber Geothermal Field. a** 3D view of realistic injection and extraction well trajectories. Open-hole sections for geothermal operation are marked with bold colored line. Two gray planes are used to distinguish the three regions (west, central, and east) of

the study area. **b** Temporal variation of injection and production fluid temperatures in the west, central and east regions of the HGF. Each dot in the four panels represents the monthly temperature record of one well. The colors of the dots are consistent with those of the open-hole sections of corresponding wells seen in **a**.

temperatures of fluid injected into their respective wells increased by ~5 °C after 2005 and decreased by ~27 °C after mid-2006, respectively. The temperature history in the central region is more complicated because all forms of operations are active: extraction, injection, and extraction later converted to injection. The temperatures of fluid extracted from 11 deviated extraction wells near the feeder fault were ~188 °C in 1985, declined sharply by ~55 °C in 1986, and then increased to ~150 °C in 1988; from 1988 to 2017, the average fluid temperature decreased by ~21 °C. In 1993, 20 vertical extraction wells began large-scale heat production with fluid temperatures of ~174 °C, which decreased to ~161 °C by 2017. At the deviated wells near the feeder fault, temperature decreased at approximately 0.72 °C/yr, which is ~33% higher than at the vertical wells (0.54 °C/yr). Such irregular temperature variations indicate that thermal effects within the reservoir are inhomogeneous in space and time.

To resolve the physical mechanisms of HGF deformation and quantify the spatiotemporal evolution of their strengths, we compile and analyze more than two decades of geodetic and seismic observations. We construct a realistic 3D thermo-hydro-mechanical model of the reservoir and bounding faults to simulate the full operation history of the HGF over 32 years, with results constrained by the surface displacement observations and reported geothermal brine temperatures (see Data Availability). Here we show that the pressure effects of pressure fluctuation and poroelastic responses keep relatively stable during the operation history, but the strength of thermal contraction associated with the reinjection of cooled geothermal fluids increases gradually with time and eventually overwhelms the pressure effects. Long-term trends of surface displacement and seismicity growth at the HGF are dominated by thermal contraction, while the pressure effects drive their short-term changes.

## Results

### Decade-long surface displacement anomalies

Vertical and horizontal ground displacements at the HGF have been measured by leveling and interferometric synthetic aperture radar (InSAR) techniques with relatively high temporal and spatial resolution. Leveling surveys with a dense network of benchmarks have been conducted approximately annually since 1994[23] (Supplementary Fig. 1). The displacement time series of each benchmark are used to characterize the vertical displacement trends of the HGF and also three subregions: west, central and east. Also, we use the time series to generate vertical displacement rate maps of four time periods: January 1994 to December 2004, February 2006 to November 2010, November 2010 to December 2014, and December 2014 to January 2018 (Supplementary Fig. 2), with the Kriging method[24]. InSAR monitoring is implemented with the Envisat satellite images (May 2006 to September 2010) and the Sentinel-1 satellite images (April 2015 to December 2017, June 2018 to August 2019) based on the SqueeSAR approach[25]. More details on geodetic data processing are provided in the Methods section.

The leveling displacement maps and time series show that the central region of the HGF experienced long-term subsidence with the maximum rates increasing from ~1.5 cm/yr to ~4 cm/yr in 2005 (Fig. 3). In contrast, both west and east regions are characterized by reversals of vertical displacements around 2005. The west region was subsiding at rates up to 1.5 cm/yr from 1994 to 2005 and then transitioned to uplift with rates up to 2 cm/yr. The east region shows a reversal in displacement trends: initially uplifting at rates up to 0.5 cm/yr and then subsiding with rates up to 2 cm/yr. The displacement rate and trend changes that initiated around 2005 and lasted for more than 10 years are referred to as decade-long transients. The leveling-observed transients are confirmed by InSAR measurements, but the leveling-measured subsidence at the HGF center is significantly larger than the InSAR result (Fig. 3b), which is most likely due to differences in the resolution and error sources between the two independent methods. Moreover, the InSAR-derived horizontal displacement maps of 2006-2010 and 2018-2019 reveal the central region was moving eastward with rates up to 2 cm/yr (Supplementary Fig. 3).

To investigate potential relationships between surface displacement and geothermal operations, we compare vertical displacement trends of the west, central, and east regions of the HGF with the respective injection and extraction volume changes of geothermal

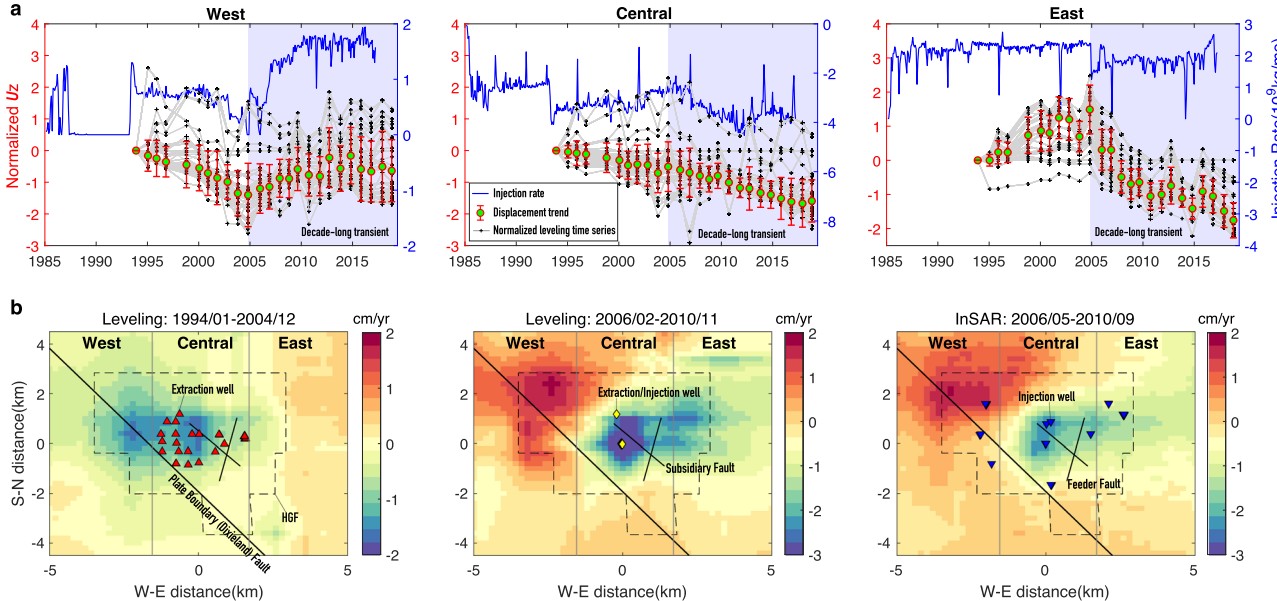

**Fig. 3 | Observed vertical displacement trends of the Heber Geothermal Field. a** Comparison of normalized vertical displacement (*Uz*) trends (red bars) of the west, central and east regions with injection rates shown by blue curves. The red bars reflect the variability in normalized leveling time series rather than observation errors. Gray curves with cross symbols show the normalized displacement time series of leveling benchmarks (see also Supp. Figure 1). The displacement trends are obtained through averaging the normalized time series. Shaded regions mark decade-long displacement transients. The blue curves with axes on the right show net injection rate for wells in three sub-regions: positive and negative values represent net injection and net extraction, respectively. **b** Vertical displacement rate maps derived from interpolation of leveling and InSAR observations in three time periods.

fluid (Fig. 3a). In the west region, with only fluid injection (Fig. 2a and Supplementary Fig. 4), the vertical displacement is characterized by subsidence before 2005 and uplift after 2005 with temporal variations which are similar to the changes of injection rates. It is common that fluid injection causes surface uplift due to pore pressure increases within the target reservoir[26,27]. However, the ground of the west region was subsiding from 1994 to 2005 even with a net increase of injected fluid, which is an unexpected phenomenon. The same displacement anomaly of ground subsiding with a net increase of injected fluid is also seen in the east region after 2005. In the central region with net fluid extraction since early 1985, subsidence was relatively constant until the post-2005 transient of acceleration, which corresponds to the prominent increase of production rates after 2005. The comparisons above indicate that reservoir pressure depletion associated with fluid loss is not the sole driver of ground deformation.

## Seismicity rate growth

Geothermal field deformation refers to not only surface displacement but also induced earthquakes. Low levels of seismicity at the HGF were first detected by Southern California Seismic Network (SCSN) in 1988, three years after the first power plant came online. Since then, there have been multiple episodes of transient and rapid increases in seismicity rates superimposed on a steady background rate (Fig. 4). The most notable one started in 2005 and lasted for ~2 years. To verify the temporal growth of seismicity from the SCSN catalog, we further use the method of template matching to improve the temporal resolution of the regional earthquake catalog and account for variations in detection levels due to network changes (see Methods for more details). From 2001 to 2018, 487 earthquakes are detected within the HGF, including twelve 2 < M < 3 and one M > 3 events, about eight-fold larger than the count of the SCSN catalog. The template matched catalog confirms the increasing trend of seismicity with time. In

addition, we find two interesting phenomena: (1) rapid increases of earthquake within the template-matched catalog occurred at the time with prominent changes of both injection and production rates; (2) the temporal growth of seismicity rates is similar to the vertical displacement trend of the HGF shown by the normalized leveling time-series, suggesting that the physical mechanisms driving surface displacement also control the occurrence of earthquakes.

## Fault slip modeling

The HGF is located within an active plate boundary zone, which implies a possibility of ground displacement associated with fault slip. More specifically, the strike-slip faults, which bound the HGF laterally could represent active, blind extensions of the Superstition Hills fault to the north (Fig. 1). As the surface displacement at the HGF consists of two parts—long-term displacement initiated before 2005 and transient displacement after 2005 (Fig. 3)—we thus conduct three scenarios of fault slip modeling with available displacement observations. First, the long-term vertical displacement measured by leveling from 1994 to 2004 is inverted with slip on the normal-slip feeder fault. Second, the vertical transient displacement from 2006 to 2010, obtained through removing the long-term displacement trend, is also inverted with the feeder fault. Third, the horizontal displacement measured by InSAR from 2006 to 2010 is modelled with the strike-slip plate boundary (Dixieland) fault. More details on inversions for fault slip refer to Methods.

Our inversions show that the magnitudes of normal slip on the feeder fault are required to be 0.4 to 1 m above the depth of 12 km to fit the long-term vertical displacement (Supplementary Fig. 6), which indicates slip rates up to 4 to 10 cm/yr. In contrast, the slip rates of the northwest extension of the nearby plate boundary fault are less than 10 mm/yr[17–19], much smaller than the inversion result. On the other hand, such large cumulative slip would cause catastrophic, irreversible

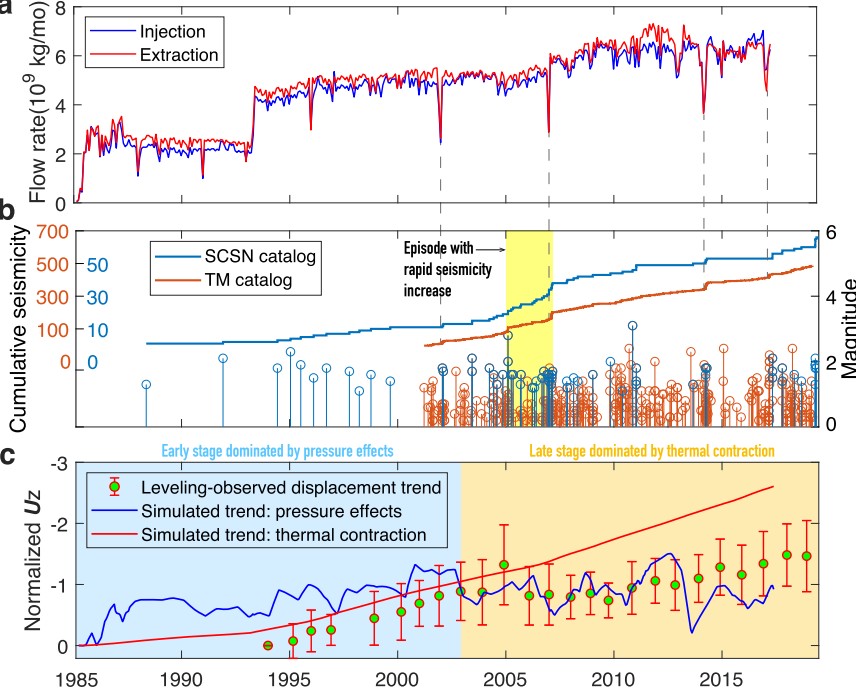

**Fig. 4 | Temporal distribution of operation volumes, seismicity and vertical displacement at the HGF. a** Monthly injection and extraction volumes for all HGF wells. Production rates of each well refer to Supplementary Fig. 5. **b** Temporal distribution of earthquake magnitudes and cumulative seismicity within the scope of the HGF based on the regional (SCSN) and template matching (TM) catalogs. The vertical dashed lines denote periods of rapid seismicity rate changes associated with rapid decreases in both the fluid injection and extraction volumes. **c** Vertical

displacement (**Uz**) trends of the HGF. Red bars show the normalized displacement trend from leveling time series. Blue and red curves represent the simulated displacement trends caused by the pressure effects (pressure fluctuation and poroelastic responses) and thermal contraction effect, respectively. The simulated trends are obtained from the normalization of surface displacements at 8 probe points (see Fig. 7 for their locations). Note the vertical scale is inverted such that an increasing trend represents increasing subsidence.

casing deformation of the wells intersecting with the feeder fault; but there were no related reports on such damage, and production rates have remained relatively constant. Therefore, it is implausible that the feeder fault was slipping at the prominent rate. For the transient vertical displacement characterized by not only subsidence but also uplift, major displacement features cannot be fit by normal slip on the feeder fault (Supplementary Fig. 7). To match the InSAR-measured horizontal displacement from 2006 to 2010, the magnitudes of dextral strike-slip on the plate boundary fault need to be up to 30 to 82 cm below the depth of 8 km (Supplementary Fig. 8), which suggests 6 to 16 cm/yr of slip rates, much higher than the reported slip rates and the relative plate motion rate (only ~4 cm/yr)[28,29]. Consequently, the dominant reason for the vertical and horizontal displacement is unlikely related to fault slip.

## Thermo-hydro-mechanical (THM) modeling

To resolve the deformation mechanisms, we further investigate the effects of geothermal operations through simulating the injection and extraction history of the HGF with a 3D coupled thermo-hydro-mechanical (THM) model (Supplementary Fig. 9). Geological structures and geothermal gradients of the geothermal system are determined according to previous studies[16,21,30–32] (Fig. 1b) and seismic velocity profiles of the study region from the Southern California Earthquake Center Community Velocity Model-Harvard (CVM-H)[33] (Supplementary Fig. 10). Variations in rock properties with depth are relatively well resolved in the CVM-H but there is little-to-no geophysical evidence of lateral variations on scales smaller than the HGF. Thus, to honor the natural complexity of the system while keeping the problem numerically tractable, we simplify it with five layers: caprock at the surface, two reservoir layers, a basal layer, and basement rock. Additionally, we embed the two NW-trending strike-slip faults and the NE-trending feeder fault in our model as individual formations with a unique set of thermo-hydro-mechanical properties.

The model layers and faults are assumed to behave as linear thermo-poro-elastic media. The density, Young's modulus, and Poisson's ratio of each layer are calculated based on the local seismic velocity structure. The corresponding mechanical properties of the three faults are set with typical values[34–36]: 2500 kg/m³, 10 GPa, and 0.3. For the thermophysical properties, we set the specific heat and thermal expansion coefficients to be 1000 J/(kg·°C) and $10^{-5}$ °C$^{-1}$, respectively, which are typical values for rocks[37]. The thermal conductivities of the two reservoir layers, basal formation, and feeder fault are set to be two-fold larger than the values of the other formations (i.e., caprock, basement, and the two strike-slip faults). Another critical aspect of a THM reservoir model is the permeability structure. According to published permeability models of the HGF[16,30], the two reservoir layers are characterized with relatively high permeability, and the feeder fault serves as the dominant upward channel of deep hot geothermal brine. Moreover, given that the strike-slip faults are part of an active plate boundary system, it is plausible that they have damage zones with enhanced permeability[38]. We thus prescribe finite widths and relatively high permeabilities to the faults.

The mechanical properties of the five layers are well constrained by the seismic velocity profiles, but all other properties are assigned based on qualitative models or prior data from general earth materials, and thus have some uncertainties. Due to the intercoupling of dynamic and nonlinear physical processes within the geothermal system, inverting for model parameters is computationally infeasible. Therefore, we calibrate the uncertain parameters through an iterative procedure with a subset of surface displacement observations and fluid temperature records. As hydrogeological properties of the reservoir have a major influence on surface displacement[39,40], we first calibrate its porosity, permeability, Biot coefficient, and three thermophysical parameters (specific heat, thermal conductivity and expansion coefficient) successively. Second, we calibrate the same parameters of the

three faults as above, including their thicknesses. Third, the heat flow rate within the feeder fault was estimated mainly based on the initial temperature distribution prior to production[30–32]. In our starting model, only the feeder fault is assumed to be the upwelling conduit of deep heat flux. To identify the replenishment mechanism of heat flow at the HGF, we conduct additional simulations under two scenarios: (I) one upwelling channel of the feeder fault with different rates of heat flux; (II) two channels of the feeder and subsidiary strike-slip faults with different flow rates. Lastly, we test the influences of different fluid properties, including density, viscosity, compressibility, thermal conductivity, and specific heat, which were initially assigned to be 1000 kg/m³, $2.5 \times 10^{-4}$ Pa·s, $4.2 \times 10^{-10}$ Pa$^{-1}$, 0.68 W/(m·°C), and 4200 J/(kg·°C), respectively.

For each parameter combination, we simulate the operation of the HGF from Apr 1985 through Dec 2010; the simulation results are compared with the vertical displacement time series (Jan 1994 - Nov 2010) at 64 leveling benchmarks, InSAR displacement maps in both vertical and horizontal directions (May 2006 to Sep 2010), and temperature records of 16 injection and 16 extraction wells. The benchmarks and wells are selected to give approximately uniform coverage within the HGF (Supplementary Fig. 1 and 4). The optimal parameter values are identified based on the misfits of simulation results to displacement and temperature observations. As the observation period of the leveling data is four-fold longer than the InSAR observation period, we set their weight ratio to be 4:1 to calculate the misfits of surface displacement. To minimize potential uncertainties of the optimal solution associated with the choices of test parameter values, we conduct two rounds of model calibration (more details refer to Methods).

After calibration, only the values of three reservoir parameters are changed relative to initial settings (Supplementary Tables 2-14). First, the permeability of the lower reservoir is decreased from 20 mD ($2 \times 10^{-14}$ m²) to 10 mD. Although the upper and lower reservoirs are characterized with the same porosity of 18%, the upper reservoir (20 mD) is more permeable than the lower reservoir. The porosity and permeability of the feeder and subsidiary faults are equal to the values of the upper reservoir. We test whether the reservoir layers and the three faults have different permeabilities in the horizontal and vertical directions, but simulation results reveal that they deform like isotropic porous media. Second, the thermal expansion coefficients and conductivities of the two reservoir layers are increased from $1.0 \times 10^{-5}$ °C$^{-1}$ to $1.3 \times 10^{-5}$ °C$^{-1}$ and from 2 W/(m·°C) to 3 W/(m·°C), respectively, higher than the corresponding parameter values of the other formations. The reservoir is thus characterized with faster heat conduction and a more prominent deformation response to temperature changes. On the other hand, our simulations show that the fits to displacement and temperature observations reduce with increasing rates of heat flow upwelling through the feeder fault, and that the root-mean-squared (RMS) misfits of displacement become larger when the subsidiary strike-slip fault is included as another upward channel of heat flux (Supplementary Table 15). Therefore, the feeder fault is the major heat flow channel of the HGF, with replenishment rates unlikely exceeding ~20 kg/s. Moreover, the simulations with different fluid properties show that the model misfits are lowest with the initial parameter settings (Supplementary Table 16).

The model calibration also provides insights into the sensitivities of simulation results to changes in the rock and fluid properties. We quantify the sensitivities with the variation of RMS misfits to displacement and temperature observations (Fig. 5). Overall, the simulated surface displacements are more sensitive to the parameter changes than the temperature. Among all test parameters, the influences of hydraulic properties and fault thickness are largest, indicating that surface displacement and heat transfer of the HGF are mainly associated with fluid flow. Beyond those, the variation of RMS misfits to displacement observations are up to ~16-17% with test thermal

expansion coefficients and specific heat of the reservoir, which is an indicator of a non-negligible thermoelastic effect. The temperature is most sensitive to the permeability changes of the feeder fault that serves as the replenishment channel of heat flux. For the fluid properties, the density, viscosity, compressibility, and specific heat all have a prominent influence on the simulation results for surface displacement.

Our parameter calibration improves the model fit to the displacement and temperature observations. The weighted RMS misfits of displacement are reduced from 2.89 cm to 2.45 cm; the RMS misfits

of temperature are decreased from 21.55 °C to 20.3 °C (Supplementary Tables 4-16), respectively, with major improvement from the permeability optimization of the lower reservoir and the optimization of the thermal expansion coefficients and conductivities of the two reservoir layers. We further use the calibrated model to simulate the operation of the HGF from Apr 1985 to May 2017, ~6.5 years longer than the time span of model calibration, with available production data. Major features of observed surface displacement in both vertical and horizontal directions can be replicated with our model (Fig. 6 and Supplementary Fig. 13). After 32 years of operation, the largest surface subsidence at

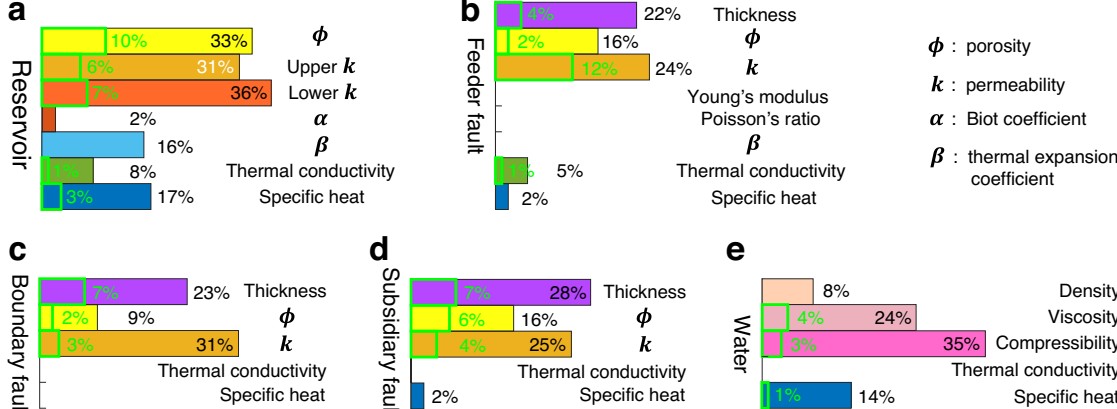

**Fig. 5 | Model sensitivities to test parameter values.** Variations of RMS misfits to displacement and temperature observations with **a** different hydraulic and thermophysical parameters of the two reservoir layers, **b** different parameters of the feeder fault, **c** different parameters of the boundary fault, **d** different parameters of

the subsidiary fault, and **e** different fluid parameters. Multicolor and green rectangles quantify the misfit variations to displacement and temperature observations, respectively. Color gaps indicate no variations.

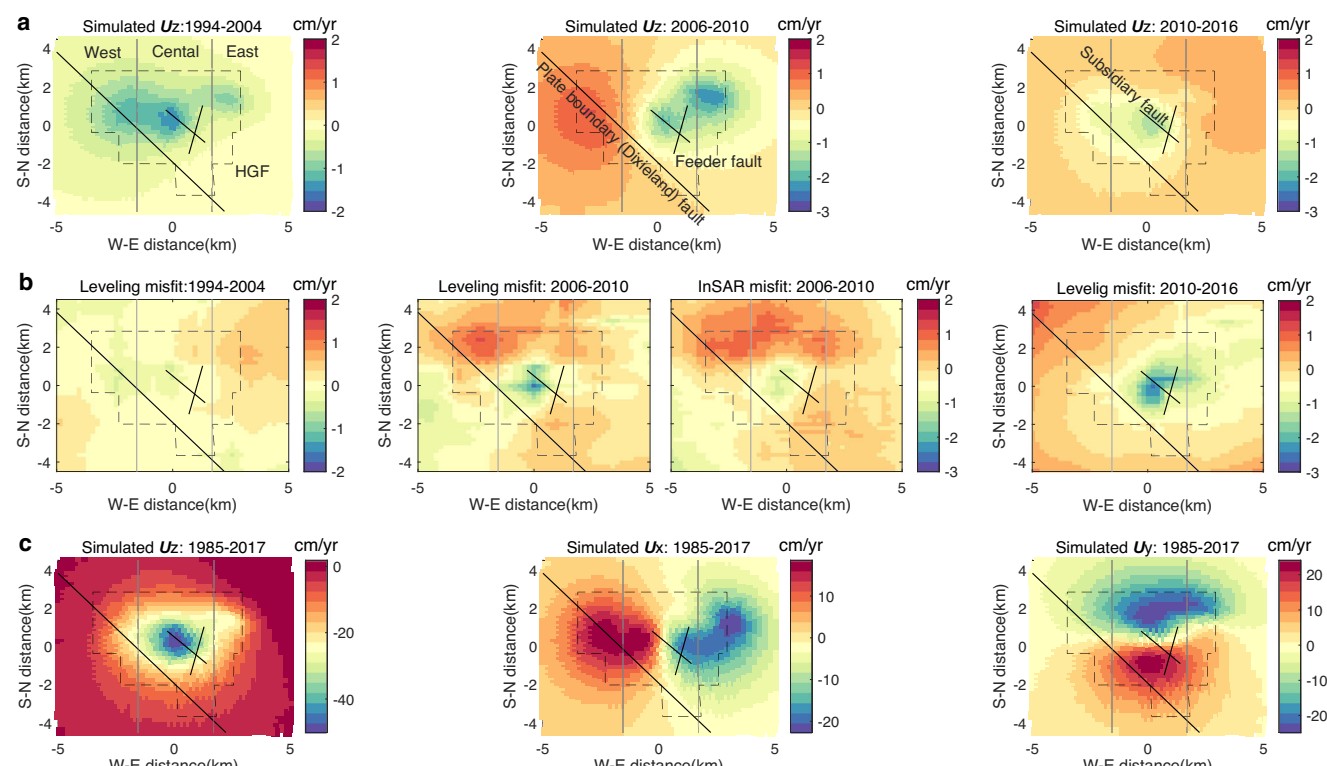

**Fig. 6 | Model predictions to surface displacement observations. a** Simulated vertical displacement ($U_z$) in 3 time periods. **b** Model misfits to leveling and InSAR observations shown in Fig. 3b and Supplementary Fig. 2. **c** Simulated surface

displacements in vertical and horizontal ($U_x$, $U_y$) directions over the full simulation period. Uplift, eastward and northward motions are defined to be positive.

the HGF center is up to 50 cm. In addition, there is also significant horizontal displacement towards the center with the maximum magnitude up to ~30 cm. However, the prominent leveling subsidence at the HGF center after 2010 and the observations in the peripheral region of the HGF are not reproduced well by our model (Fig. 3b). One possible reason for the difference between the simulated and observed displacements is that the model parameters calibrated with available observations represent the rock properties in average and cannot reflect laterally inhomogeneous hydromechanical and thermophysical properties and their variations in the depth to basement rock of the study region[41]. The effect of inhomogeneous rock properties increases with the duration of geothermal simulation, as shown by the decreasing goodness-of-fit between the model predictions and the leveling time series at the HGF center (Supplementary Fig. 14). Another possible reason is that the displacement observations in the peripheral region of the HGF, where no geothermal operations occur, may be related to other physical processes like natural recharge of regional groundwater. In addition, it is worth noting that the leveling-observed subsidence at the HGF center is larger than the InSAR monitoring results (Supplementary Figs. 2 and 3).

Our calibrated model can also predict the reported temperature variations for most wells (Supplementary Fig. 16). For the six deviated extraction wells intersecting the feeder fault (Fig. 2a), there are systematic biases between the model predictions and observations, which could indicate a mismatch between actual temperature distribution in 1985 and the initial temperature conditions used in the model (Supplementary Fig. 11). In addition, our model cannot reproduce ~30 °C temperature decreases of 3 selected vertical extraction wells (identifiers 2591216, 2591217, and 2591221) after 2000, which are likely due to inhomogeneous hydraulic properties of the reservoir. Strong heat convection associated with fluid flow may occur in some localized

regions due to higher porosity and permeability than the calibrated values.

## Reservoir pressure and temperature evolution

Our simulations show that the movement of fluid within the reservoir is controlled to a first order by flow from injection wells towards extraction wells (Fig. 7a). Pore pressure variations are widespread within the reservoir, but temperature decreases are only seen in localized regions around injection wells. To inspect the spatiotemporal changes of reservoir pressure and temperature, we inspect the model results at eight probe points across the model domain located at the bottom of the upper reservoir (1.5 km depth). Pressure time series at these points show that dramatic fluctuations mainly occurred within four periods: 1985-1988, 1992-1994, 2003-2006, and 2012-2015 (Fig. 7c). The maximum pressure fluctuation – pressure decrease – occurred near the feeder and subsidiary faults from 1985 to 1988, dropping approximately 3 MPa from the initial pressure of 14.7 MPa to ~11.7 MPa (points P5 and P6). In contrast, pressure within the plate boundary fault (point P3) was decreasing slowly before 2006 by ~1.5 MPa relative to its initial value, and then started increasing but at a lower rate.

In the western region of the HGF, although the net injection rates are around $0.6 \times 10^9$ kg per month on average from 1994 to 2005, pressure decreased by 0.5-1.0 MPa (points P1-P3). From the Darcy velocity field (Fig. 7a), we can deduce that the pressure decrease is attributable to the loss of fluid that was flowing toward the center. High production rates around $3.5 \times 10^9$ kg/mo at the HGF center created a low-pressure zone siphoning fluid from surrounding regions. After 2005 the average injection rates increased up to ~$1.6 \times 10^9$ kg/mo, and the pressure rose gradually, indicating that the injection volume became larger than the loss. For the central region, probe points P4-P6

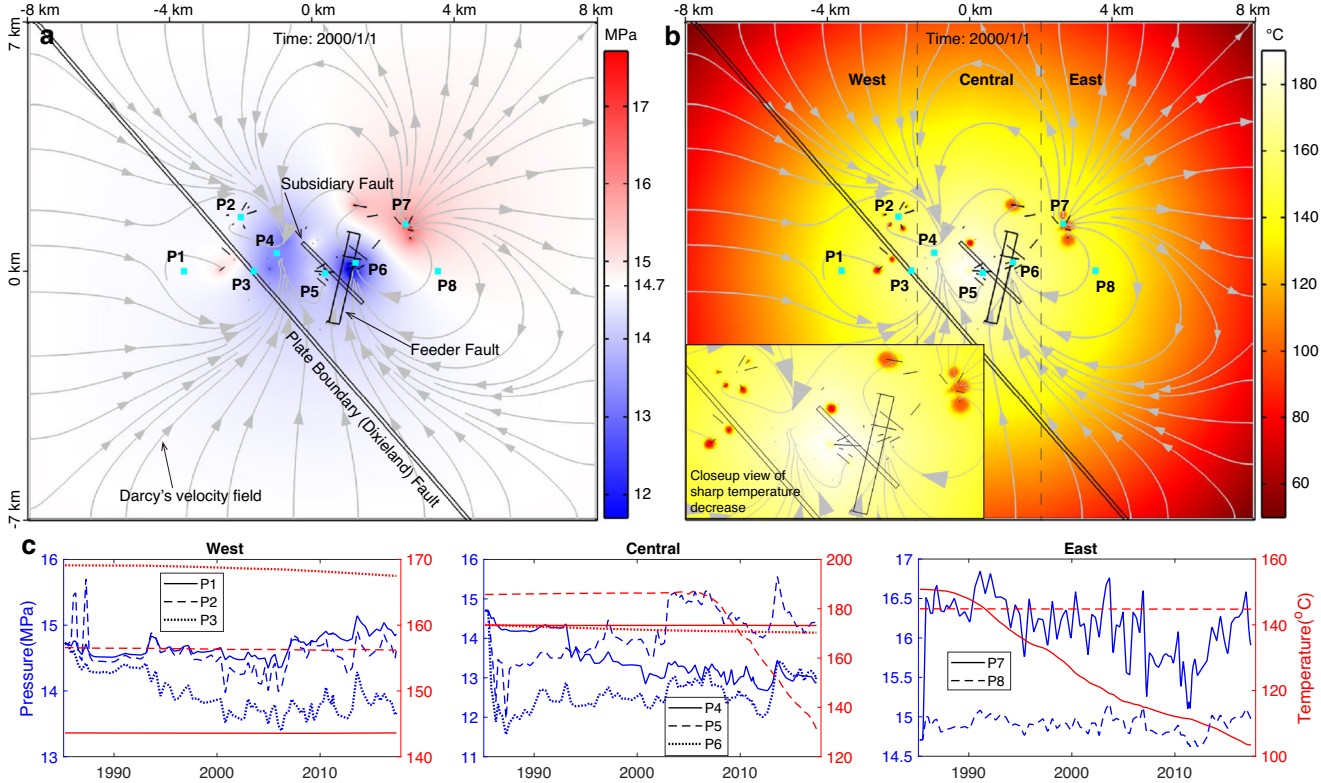

**Fig. 7 | Simulated pressure and temperature changes at the bottom of the upper reservoir (depth = 1.5 km). a** Pressure distribution at a time node after ~8 years of geothermal operation with gray curves and arrows showing the movement of fluid. Red and blue colors represent an increase and decrease in pressure relative

to the initial pressure (14.7 MPa), respectively. **b** Temperature distribution after ~8 years of operation. **c** Pressure and temperature variations at 8 probe points distributed within the west, central and east regions of the HGF.

show different trends: pressures declined continuously by ~2 MPa among the vertical extraction wells (point P4) but declined abruptly near the feeder and subsidiary faults (points P5-P6) during the first three years of operation. The rapid decline is likely due to exhaustive exploitation, which is probably the major reason why the operator decommissioned the binary plant in 1987. To boost power generation capacity, 9 production wells were closed in 1988 or 1989 and then converted to injection around 1993. In the east region with average injection rates of ~2.1×10⁹ kg/mo, the pressure at locations within and ~1.5 km far from the well cluster (points P7 and P8) fluctuates around average values of ~16 MPa and ~14.9 MPa, respectively, across the whole time period.

In contrast to the sharp changes of pore pressure, fluid temperature varies smoothly over time (Fig. 7c), indicating insensitivity to abrupt changes in production rates and slower heat transfer than pressure diffusion. In addition, the sizes of localized temperature anomalies scale with injection rates. For instance, around probe points P5 and P7, the areas with temperature decrease over 20°C eventually merge due to dense injection wells and large injection volumes (Supplementary Fig. 17). These two findings reveal that heat transfer within the reservoir is mainly through the convection of fluid flow driven by geothermal operations, which is supported by model sensitivity tests showing that well temperatures are more sensitive to changes in hydraulic properties rather than the thermophysical properties.

## Quantification of inhomogeneous thermo-poro-elastic effects

As the calibrated model can simulate major features of observed surface displacements and fluid temperatures, we further use it to quantify the relative influence and spatiotemporal evolution of pore pressure changes, poro- and thermo-elastic effects. The geothermal operation history of the HGF is simulated under three end-member scenarios: (I) the Biot coefficients of the two reservoirs and the three faults are set to be zero; (II) the thermal expansion coefficients of the

reservoirs and faults are assigned to be zero; and (III) both kinds of parameters are set to be zero. The effect of pressure changes can be nullified through comparing the simulate results of model I with those of the calibrated (reference) model. The thermoelastic effect can be isolated through comparison of the reference model with model II. The poroelastic effect of contraction and expansion can be obtained directly from model III.

The pressure changes within the HGF reservoir cause both subsidence and uplift, mainly in the central and northeast regions (Fig. 8a), respectively, with magnitudes varying slightly during the operation history of the HGF (Supplementary Fig. 21). By May 2017, the maximum subsidence and uplift are ~8 cm and ~4 cm, respectively. The poroelastic effects of contraction and expansion produce similar spatiotemporal features of vertical displacements as the pressure changes, but with magnitudes larger by ~20%. In contrast, the thermoelastic effect only causes ground subsidence; we thus call it thermal contraction. The subsidence is initially localized in the northeast region but gradually extends to the central and west regions (Supplementary Fig. 23). Around 2000, the subsidence in the central region became greater than the other regions. The effect of thermal contraction has been increasing steadily with time since the operation began. After only ~2 years of operation, vertical displacement at the northeast corner caused by thermal contraction exceeds the displacement associated with the joint effect of pressure fluctuation and poroelastic responses (Fig. 8b). By 2017, the thermoelastic subsidence was more than twice as large as the displacements caused by the two pressure effects.

To further compare the differences of pressure effects and thermal contraction, we inspect their temporal variations at four specific locations (Fig. 8c, 8d). In contrast to the instantaneous response and proportional relation of poroelastic responses to pressure fluctuation, thermoelastic effect varies nonlinearly and asynchronously with temperature. For example, at point P1 the ground experienced

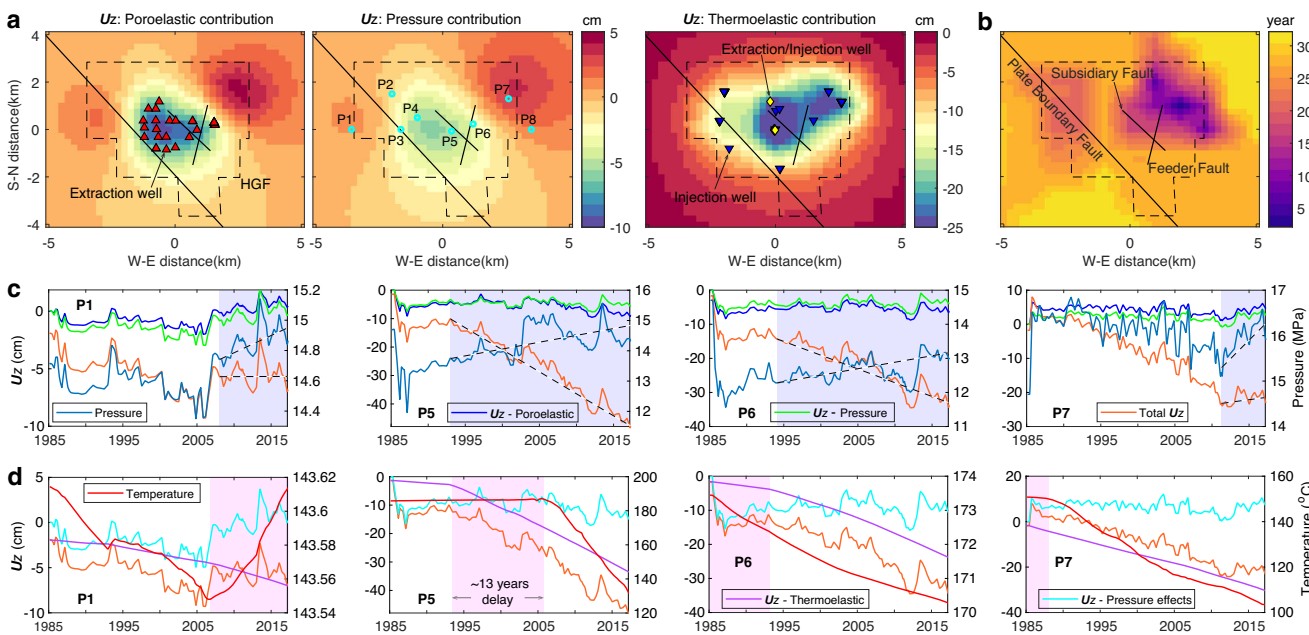

**Fig. 8 | Spatiotemporal evolution of thermo-poro-elastic effects. a** Vertical surface displacement (*Uz*) on 2016/06/17 due to pore pressure fluctuation, poroelastic responses, and thermal contraction. **b** Timing (relative to 1985/4/1) of the thermal contraction effect exceeding the joint effects of pressure fluctuation and poroelastic responses. **c** Temporal evolution of the effects of pressure fluctuation and poroelastic responses at four probe points. Shaded regions show the time periods with contrary trends (marked with black dashed lines) of cumulative vertical surface displacements (orange curves) and pressure changes (denim blue

curves) at the four points with depths equal to 1.5 km. Blue and green curves show the *Uz* changes due to pressure fluctuation and poroelastic responses, respectively. **d** Temporal evolution of the thermal effect at four probe points. Shaded regions show the time periods with opposing trends of cumulative vertical displacement and temperature (red curves). Magenta and cyan curves show the *Uz* changes due to thermal contraction and the joint effects of pressure fluctuation and poroelastic responses, respectively. Temporal evolution of thermo-poro-elastic effects at the other four points (P2-P4 and P8) refers to Supplementary Fig. 24.

thermoelastic subsidence with temperature increase during some time periods; at point P5, the thermal contraction effect increased ~13 years earlier than temperature decrease. These phenomena also demonstrate that the thermal effect can disturb broad regions beyond localized areas around injection wells with temperature decrease[42,43]. The strength of thermal contraction at a specific location not only depends on the amplitude of temperature decrease and its distance off the temperature decrease front but also the injection volume of heat-depleted brines. We can see that thermoelastic subsidence at point P6 with ~3 °C of temperature decrease is comparable to the value at point P7 with 53.5 °C of temperature decrease. Moreover, these probe points reveal that abrupt changes of vertical displacements are associated with reservoir pressure changes, which respond quickly to geothermal operations.

### Physical mechanisms of HGF deformation

Although the three mechanisms of pressure fluctuation, poroelastic responses and thermal contraction all contribute to the deformation of the HGF, their effects vary significantly in space and time. In the west region, the ground subsidence anomaly from 1995 to 2005 is jointly controlled by the three mechanisms with comparable strengths (point P1 in Fig. 8c, d). The pressure decrease was associated with the siphoning effect of the HGF center with a pressure drop of up to 3 MPa relative to the initial pressure (Fig. 8a). After 2005 with the injection volume exceeding the loss, the reservoir pressure begun to increase, which is the reason for the decade-long displacement transient of turning to surface uplift. For the central region, the transient acceleration in subsidence after 2005 is mainly caused by thermoelastic contraction, which is also responsible for the displacement anomaly of turning to subsidence around 2005 in the east region (Fig. 8c, d).

In addition, the simulated displacement time series at 8 probe points show that the short-term changes and long-term trend of vertical surface displacements are associated with the two pressure effects and thermal contraction (Fig. 8 and Supplementary Fig. 24), respectively. To identify the physical mechanisms controlling deformation of the whole study region, we further normalize the simulated vertical displacements at 8 probe points caused by the pressure and thermal effects. Leveling-observed surface displacement trend and temporal growth of seismicity within the HGF is highly correlated with the normalized vertical displacements linked to thermal contraction (Fig. 4c), with the correlation coefficients up to 0.85 and 0.97, respectively. In contrast, the coefficients of the leveling displacements and seismicity growth relative to the normalized displacements associated with the pressure effects are only 0.09 and 0.23, respectively. We thus conclude that thermal contraction dominates the long-term trend of HGF deformation including surface displacement and seismicity, while the pressure effects drive short-term changes.

## Discussion

Although the pressure and thermal effects have been recognized as the causal mechanisms of geothermal field deformation[5–11], the spatial and temporal evolution of their strengths has not been well understood. One major reason for this knowledge gap is due to the rarity of long-term seismic and geodetic observations combined with the difficulty of setting up a realistic 3D thermo-hydro-geological model with limited subsurface geological and geophysical data (like reservoir geometry, formation lithology and fault distribution), which are essential to simulate decades of geothermal operations. Our framework, developed for resolving the spatiotemporal evolution of the HGF deformation mechanisms through integrating multiple geodetic, geophysical and geological data, is applicable to other complex geothermal systems.

One benefit of having a realistic 3D THM model is the ability to identify distinct features of pressure and thermal effects. At the HGF we gain the following insights: (1) the pressure effects respond much

faster to changes in geothermal operations than the thermal effect, (2) in contrast to the instantaneous response and proportional relation of poroelastic effects to pressure fluctuation, the effect of thermal contraction varies nonlinearly and asynchronously with temperature; and (3) the pressure effects are relatively stable during geothermal operations whereas the thermal effect increases gradually with time and eventually overwhelms the pressure effects. Therefore, the pressure and thermal effects dominate the early- and late-stage deformation of geothermal fields, respectively. This finding helps to reconcile the controversy of differing deformation mechanisms reported for the same geothermal field, like at the Coso field[5,10] and the Brady Hot Springs field[9,44].

While the thermal effect has been recognized for its role of controlling the geomechanical behavior of enhanced supercritical geothermal systems[45], and inducing earthquakes near reinjection wells with localized temperature drop[36] and in regions over the temperature decrease front[42,43], here we show that thermal contraction dominates long-term deformation trend of the whole Heber geothermal field including seismicity growth and surface displacement. In contrast, the pressure effects of pressure fluctuation and poroelastic responses drive short-term deformation changes. These findings provide theoretical implications for mitigating geohazards at geothermal fields and for improving the sustainability and safety of geothermal operations.

## Methods

### Leveling data analysis

There are 158 leveling stations including a reference benchmark (A-33) at the HGF (Supplementary Fig. 1). By the end of 2018 these benchmarks were surveyed 25 times: January 1994, March 1995, January 1996, December 1996, December 1998, March 2000, January 2001, December 2001, December 2002, December 2003, December 2004, February 2006, January 2007, January 2008, December 2008, October 2009, November 2010, December 2011, November 2012, December 2013, December 2014, December 2015, December 2016, January 2018, and December 2018. However, only 103 stations were included during every survey. The observation errors of leveling range from 0.2 cm to 0.5 cm. To characterize vertical displacement trends of the HGF, we divide the study area into three regions (west, central, and east) to analyze the leveling observations. The displacement time series of the 103 stations are first normalized by their root-mean-square values with the following expression,

$$n(i) = u(i) \Big/ \sqrt{\sum_{i=1}^{N} \frac{u(i)^2}{N-1}} \qquad (1)$$

where $u(i)$ and $n(i)$ are the measured and normalized displacement, respectively, at each time node; and $N$ is the total number of surveys. We then average the normalized time series of the stations within each sub-region to represent its displacement trend. For the whole study area with both surface uplift and subsidence, absolute values of all normalized time series are averaged to represent the overall trend of surface displacement.

The time series of leveling displacements are also used to generate vertical displacement rate maps of four time periods: 1994/01-2004/12, 2006/02-2010/11, 2010/11-2014/12, 2014/12-2018/01, which are divided based on the observed displacement rate changes (Supplementary Fig. 1) and time periods of InSAR data (Supplementary Fig. 3). We first calculate the displacement rates of the stations with at least four surveys during each time period (left column in Supplementary Fig. 2), and then derive the maps (right column in Supplementary Fig. 2) through interpolating the discrete points with the Kriging method[24].

## InSAR data processing

Surface displacement of the study area is also measured with SAR images of the Envisat and Sentinel-1 satellites. All the data are processed with the SqueeSAR approach[25], which utilizes permanent and distributed scatterers to measure the surface displacement in the line-of-sight (LOS) direction to the satellites. The LOS displacement rate maps of ascending and descending tracks of each time period are further decomposed into vertical and east horizontal components using the approach proposed by Eneva et al.[23] (see Supplementary Fig. 3). Although most scatterers are aligned along roads and canals, their spatial coverage is dense enough to detect displacement features of the HGF. We also adopt the Kriging method[24] to generate complete displacement images through interpolating the scatterers for comparison with the leveling rate maps.

## Earthquake template matching

Template matching is an important observational tool for characterizing repetitive seismic sources[46]. Multi-station template matching often involves cross-correlating the waveforms that were recorded from a cataloged earthquake across a network against years of continuously recorded data to detect similar, smaller magnitude earthquakes. Here, we use all earthquakes of the SCSN catalog occurred in the Heber area from 2001 to 2018 as templates. Parameters used in template creation and event detection are consistent with those in our previous work[47]. Templates are 20 second in length and begin 5 second prior to P-phase arrivals on vertical channels and 5 second prior to S-phase arrivals on horizontal channels. All seismic waveforms are filtered between 5-15 Hz, and these templates are then cross-correlated against a five-station network of seismometers, namely MONP from network AZ, and DRE, EMS, ERP, and WES from network CI. Newly detected events have to exceed a threshold of 15 times the hourly median absolute deviation of the network normalized cross-correlation coefficients (NNCCC) in addition to the NNCCC exceeding a coefficient of 0.2.

## Fault slip inversion

First, the normal-slip feeder fault and strike-slip plate boundary fault are set to be 20 km wide and discretized with 0.46 km (long) × 2.0 km (wide) and 1.46 km (long) × 2.0 km (wide) rectangular dislocation patches, respectively. Second, we conduct linear slip inversions with the bounded variable least squares algorithm[48]. The Green's functions relating observations to slip on the patches are calculated with the elastic half-space dislocation model[49]. To add virtual observations and avoid singularities in the slip distribution, we apply the modified Laplacian operator[50], which can smooth slip on boundary patches of a finite fault reasonably, during each inversion with the smoothing factor set to be 0.1.

## Reservoir model setup—3D geometric model and tetrahedral meshing

We construct a 3D model to represent the HGF. The spatial scale of the model domain is 16 km long (E-W), 14 km wide (N-S), and 8 km deep, which is two-fold larger than the scope of the HGF reservoir (Supplementary Fig. 9). The geographical coordinates of the model center are coincident with well 2591224 (−115.535 °E, 32.712 °N). In the depth direction, based on the geological model (Fig. 1b) and seismic velocity model of the study area (Supplementary Fig. 10), we divide the model into five layers: caprock (0-0.55 km deep), upper reservoir (0.55-1.65 km deep), lower reservoir (1.65-3.15 km deep), basal layer (3.15-5.3 km deep), and basement (5.3-8 km deep). In addition, all operation wells are added into the model based on their trajectories of open-hole sections, which are mainly located within the upper reservoir. Lastly, the two NW-trending strike-slip and one NE-trending normal fault are also included in our model with the thicknesses initially set to be 100 m and 300 m, respectively. For the strike-slip faults, due to unknow

surface traces, we simplify them as two lines based on the southeast extension of the Superstition Hills fault and their locations presented by James et al.[16]. We test the influence of different model sizes on simulation results and find that larger model sizes have no obvious improvement in fitting the leveling, InSAR and temperature observations (Supplementary Table 3).

The 3D model is discretized with tetrahedral elements for finite element analysis. According to the sizes and roles (significance) of wells, faults and layers, we classify them into four categories: (1) wells, (2) two strike-slip faults, (3) caprock, feeder fault and reservoirs, (4) basal layer and basement. The minimum and maximum sizes of tetrahedral elements used to discretize wells are 3 m and 50 m, respectively. For the two strike-slip faults, the minimum and maximum sizes are 5 m and 130 m, respectively. For the third category of model domain, the minimum and maximum sizes are 7 m and 270 m, respectively. For the fourth category of model domain, the minimum and maximum sizes are 9 m and 410 m, respectively. After discretization, there are 1166303 domain elements, 77696 boundary elements, and 5012 edge elements in total. We also test other mesh schemes and find that model simulation with the preferred mesh scheme can obtain an acceptable balance between resolving large stress gradients and computational efficiency.

## Reservoir model setup—Parameter setting for mechanical, hydraulic and thermophysical properties

The mechanical parameters of Young' modulus $E$ and Poisson's ratio $\nu$ of the five layers are calculated based on the velocity and density profiles of the study area (Supplementary Fig. 10). For the faults in the model, their values of density, Young's modulus, and Poisson's ratio are initially assumed to be 2500 kg/m³, 10 GPa, and 0.3, respectively, which are widely-used values[34–36]. The Biot coefficient $\alpha$, corresponding to the effective stress coefficient for bulk deformation, is an important parameter characterizing the influence of pore pressure changes on the deformation of rock skeleton. We calculate the parameter values with two equations: $\alpha = (1 + \nu + 2\phi(1 - 2\nu))/(3(1 - \nu))$[51]; $\alpha = 1 - K/K_s'$[52]. $\phi$ is the porosity of rock. $K$ is the drained bulk modulus equal to $E/[3(1 - 2\nu)]$. $K_s'$ is the unjacketed bulk modulus, often called the solid-grain modulus, and here is assumed to be 100 GPa. Extreme values of $\alpha$ approaching 1 and 0 corresponds to unconsolidated material (or highly damaged rock) and a porosity-free aggregate[53], respectively. There is some difference between the calculation results with the two equations (Supplementary Table 2). We thus test the influence of different Biot coefficients of the reservoirs and faults on the simulation results during subsequent model calibration.

For the hydraulic parameters of porosity and permeability, we first set their starting values and then conduct model calibration with a subset of available surface displacement observations and fluid temperature records. Guided by previously published 2D permeability models[16,30], the two reservoir layers, the feeder and subsidiary faults are set with relatively high values in contrast to the other formations (Supplementary Table 2). The porosity and permeability of caprock (predominantly shales) are set to be 3% and 0.28 mD[54]. For the two reservoirs, we initially set the porosity and permeability to be 18% and 20 mD, respectively, based on two published studies[21,30]. The porosity and permeability of the subsidiary and feeder faults are initially assigned to be 20% and 20 mD, respectively, close to the values of the reservoirs. The hydraulic properties of the other formations are assigned based on the corresponding lithology. For the thermophysical properties of rocks within different formations, we set their values according to Robertson (1988)[37].

## Reservoir model setup—Initial and boundary conditions

We set two kinds of initial conditions for the model: (1) temperature distribution; (2) stress equilibrium between gravity and hydrostatic pressure. For the initial temperature, we refer to the published

temperature models[21,30–32] and assume its distribution to be axially symmetric (Supplementary Fig. 11). In the depth direction, the temperatures at the surface and the depth of 10 km below the model center are set to be 22 °C and 370 °C, respectively. The gradients of temperature increase ($\triangle T_d$) in the depth direction below the center are 55.2 °C/km and 11 °C/km, respectively, for the upper two layers. For the other three layers at the center and the model margin (10 km away from the center), the $\triangle T_d$ values are assigned to be 25 °C/km. In the horizontal direction, the temperature is assumed to decrease linearly with the radius far from the center at different depths. For the initial stress equilibrium, we exert both gravity and hydrostatic pressure on the model. However, there are minor residual stresses that cannot be balanced with this method; we thus use the simulated stress at $t = 0$ to balance the residual stress.

The boundary conditions consist of mechanical, hydraulic and thermal conditions. The mechanical boundary conditions include: (1) free upper surface, (2) roller boundaries for the bottom and side surfaces, and (3) fixed open-hole sections for all wells. The hydraulic boundary conditions comprise: (1) zero pressure at the upper surface, (2) an upwelling heat fluid source on the bottom of the feeder fault with the mass flux initially assigned to 10 kg/s[30–32], (3) hydrostatic equilibrium at the bottom and side surfaces, and (4) flux discontinuity for injection and production wells. The thermal boundary conditions include: (1) thermal insulation on the top surface, (2) a heat source on the bottom of the feeder fault with the temperature of heat flux assigned to be 320°C at the depth of 8 km, (3) heat exchange on the bottom and side surface (open boundary), and (4) heat flux boundaries for the open-hole sections of injection wells. It is worth noting that there is no conductive heat flux across the extraction wells.

## Thermo-poro-elasticity formulation

The theory of thermo-poro-elasticity is derived from momentum, mass and energy conservation[26,52,55]. The quasi-static equilibrium of the solid matrix of porous medium (with shear modulus $G$, Poisson's ratio $\upsilon$, permeability $k$ and porosity $\phi$) can be described with the Navier-Stokes equation,

$$G\nabla^2 u_i + \frac{G}{1-2\nu}\frac{\partial^2 u_k}{\partial x_i \partial x_k} = \alpha\frac{\partial p}{\partial x_i} + \beta K\frac{\partial T}{\partial x_i} - F_i \quad (2)$$

where $u_i$ is displacement component in direction $x_i$ of a Cartesian coordinate system. The quantities $p$ and $F_i$ are pore pressure and body force, respectively. The Biot-Willis coefficient $\alpha$ corresponds to the effective stress coefficient for bulk deformation, controlling the magnitude of poroelastic strain induced by fluid injection. $\beta$ is the volumetric expansion coefficient of the rock. $K$ is the drained bulk modulus of porous media.

The mass conservation equation describing laminar Darcy-type flow (with density $\rho_f$, viscosity $\eta$, and bulk modulus $K_f$) within porous rock is represented as

$$S_\varepsilon \frac{\partial p}{\partial t} - \frac{k}{\eta}\nabla^2 p = \frac{Q_m}{\rho_f} \quad (3)$$

where $Q_m$ is the mass source/sink term. The expression of the constrained specific storage coefficient $S_\varepsilon$ can be formulated as[26],

$$S_\varepsilon = \frac{\alpha}{K_s} + \phi\left(\frac{1}{K_f} - \frac{1}{K_\phi}\right) \quad (4)$$

where $\frac{1}{K_s}$ and $\frac{1}{K_\phi}$ are unjacketed bulk and pore compressibility, respectively. This formulation shows that the storage coefficient depends on not only the elastic moduli of the porous medium but also the fluid properties. In this study the porous media is assumed to be

isotropic and microscopically homogeneous, which indicates $\frac{1}{K_s}$ equal to $\frac{1}{K_\phi}$.

Heat transfer in porous media through conduction of fluid-solid mixture and convection of fluid flow obeys the law of energy conservation. Assuming local thermal equilibrium between the fluid (with specific heat $C_f$ and thermal conductivity $\kappa_f$) and rock (with density $\rho_s$, specific heat $C_s$ and thermal conductivity $\kappa_s$), and neglecting the dissipation of mechanical energy due to deformation of the solid, the energy conservation can be written as

$$\left(\phi\rho_f C_f + (1-\phi)\rho_s C_s\right)\frac{\partial T}{\partial t} + \rho_f C_f\nabla\cdot\boldsymbol{q}T - (\phi\kappa_f + (1-\phi)\kappa_s)\nabla^2 T = Q_e \quad (5)$$

where $T$ is the temperature of the fluid-solid mixture, and $Q_e$ is an external energy source. $\boldsymbol{q}$ is the Darcy flux, which can be expressed as $\rho_f\frac{k}{\eta}\nabla p$ without consideration of the gravitational potential and has been involved in the mass conservation equation.

There are three kinds of two-way coupling within the theory of thermo-poro-elasticity (Supplementary Fig. 12). However, the effects of convective heat transfer, poro- and thermo-elastic responses of skeletal rock deformation associated with pressure and temperature changes are expected to dominate over other effects like the influence of rock deformation on the temperature and pressure fields[56–58]. For computational efficiency, only the three major effects are included in our modeling. We employ COMSOL Multiphysics® (version 6.0, https://www.comsol.com/) to solve the governing equations. The average time of simulating 9400 days (from 1985/4/1 to 2010/12/30) of fluid injection and extraction is about 30 hours with our workstation (CPU: Intel Xeon Gold 6240 C 2.60 GHz; RAM: 384 GB DDR4 ECC REG 2933 MHz).

## Model parameter calibration

An iterative calibration strategy involving dozens of steps is adopted to optimize the model parameters assigned based on qualitative models or prior data from general earth materials (Supplementary Table 2). In each step only one kind of parameter is calibrated, and the optimal parameter derived from preceding step is inherited in next step. First, the two hydraulic parameters (porosity and permeability), Biot coefficient, and three thermophysical parameters (specific heat, thermal conductivity and expansion coefficient) of the two reservoir layers are calibrated successively. Second, we test different values for the thickness, elastic, hydraulic, and thermophysical parameters of the three faults. Third, we test whether both of the feeder fault and subsidiary fault serve as the upwelling channel of heat flux. Fourth, we test the influence of different properties of water on simulation results. After above simulation tests, we conduct a second round of model calibration to further optimize the parameters with the most prominent influence on simulated displacement and temperature. As part of the calibration process, we also perform sensitivity tests of surface displacement and well temperature observations on different model parameters. More details are listed below.

## 1st round calibration−Step I: Parameter calibration for the reservoir layers

First, the porosities of the two reservoir layers are calibrated simultaneously. We adopt the grid searching method to pinpoint the optimum values based on the fit of simulation results to leveling, InSAR and temperature observations. The search step is set to be 2%. When the porosities of the two reservoirs are both equal to 18%, the weighted RMS misfit of simulated surface displacement to the leveling and InSAR observations is least (Supplementary Table 4). In contrast, the RMS misfits to well temperature observations are least with the porosities of the upper and lower reservoirs equal to 20% and 18%, respectively. However, the weighted RMS misfit of displacement is

increased by 11% (3.25 cm to 2.89 cm) within the two groups of porosity, three-fold larger than the improvement of well temperature (21.55 °C to 20.71 °C). We thus select the optimal porosities of the two reservoirs to be 18%, which is the same as the initial model setting (Supplementary Table 2). Second, we calibrate the horizontal and vertical permeabilities of the upper reservoir also with the grid searching method. The search step is set to be 10 mD. The RMS misfits of displacement are least with the two permeabilities equal to 20 mD (Supplementary Table 5). In contrast, the RMS misfits of temperature are least when the horizontal and vertical permeabilities are equal to 20 mD and 30 mD, respectively. However, the RMS misfit of displacement is increased by 7% within the two groups of porosity, larger than the improvement (5%) of well temperature. We thus select the optimal permeability of the upper reservoir to be 20 mD. Third, the permeabilities of the lower reservoir are optimized with the same method as above. The misfits to displacement and temperature observations are smallest when the horizontal and vertical permeabilities both equal to 10 mD (Supplementary Table 6). Fourth, we test the influence of different Biot coefficients of the two reservoirs on the simulation results. The RMS misfits of displacement vary little with the Biot values varying from 0.52 to 1 (Supplementary Table 7). The RMS misfits of temperature keep constant with different Biot values. We thus retain the initial setting of the reservoir Biot coefficients. Lastly, we calibrate three thermophysical parameters: expansion coefficient, conductivity, and specific heat. The misfits of displacement are least with the three parameters equal to $1.3 \times 10^{-5}$ °C$^{-1}$, 3 W/(m·°C) and 1000 J/(kg·°C) (Supplementary Table 8), respectively. Simulated temperature shows no or little variations with different values of the thermophysical parameters.

Above simulations show that the displacement and temperature vary little with different Biot coefficients of the reservoir. Considering that the scale of the reservoir is much larger than the three faults, the influence of different Biot coefficients of the faults is also expected to be little. We thus do not include the parameter in subsequent model calibration.

### 1st round calibration—Step II: Parameter calibration for the three faults

We first test different values for the thickness of the feeder fault. Both the displacement and temperature misfits are smallest when the values are equal to 300 m (Supplementary Table 9). Second, we calibrate the hydraulic parameters of porosity and permeability. The displacement and temperature misfits are lowest with the two parameters equal to 20% and 20 mD (Supplementary Tables 9 and 10), respectively. Third, the two elastic parameters, Young's modulus and Poisson's ratio, are initially assigned with typical values for faults[34–36] and then optimized, but the simulation results are insensitive to changes in these parameters. Fourth, we test the influences of three thermophysical parameters: expansion coefficient, conductivity, and specific heat. The displacement and temperature misfits vary little with different thermal expansion coefficients, and are smallest with the initial settings.

As the two elastic parameters and thermal expansion coefficient of the feeder fault have little influences on simulation results, the effects of the three parameters of the two strike-slip (plate boundary and subsidiary) faults with thickness less than the feeder fault are also expected to be little. We thus only calibrate the other five parameters for the strike-slip faults: thickness, two hydraulic parameters, thermal conductivity, and specific heat (Supplementary Tables 11-14). Simulation results show that the displacement and temperature misfits are smallest with the initial settings.

### 1st round calibration—Step III: Simulation tests with different scenarios of heat flux upwelling

We test two scenarios: (I) only the feeder fault serving as the upwelling conduit of heat fluid; (II) both the feeder and subsidiary faults serving as the upwelling conduit. The rates of heat flux are also tested with a wide range from 1 kg/s to 100 kg/s. Under scenario I, the RMS misfits of displacement and temperature are smallest with flow rates less than 20 kg/s (Supplementary Table 15). Under scenario II, the minimum RMS misfit of displacement is 2.54 cm, ~4% larger than the least value of scenario I, indicating little heat flux upwelling through the subsidiary fault. The rate of heat flow within the feeder fault is probably less than 20 kg/s.

### 1st round calibration—Step IV: Simulation tests with different properties of water

We calibrate five parameters including density, viscosity, compressibility, thermal conductivity and specific heat for water. The RMS misfits of displacement and temperature vary with different values of density, viscosity, compressibility and specific heat but the parameter of thermal conductivity has no influence on simulation results (Supplementary Table 16). Our calibration shows that the initial parameters settings for the water properties are suitable for simulating geothermal operations of the HGF.

### 2nd round calibration

Due to the possibility that the optimal parameter values obtained from one step of the iterative calibration may change in later steps, we thus conduct a second round of model calibration to further optimize the parameters with prominent influences on simulation. We first recalibrate the hydraulic parameters and thermal expansion coefficients of the reservoirs (Supplementary Tables 4-6 and 8). Simulation results shows that both displacement and temperature misfits are lowest with the optimal parameter values determined from the first round of calibration, with no additional reductions in the misfits. As a consequence, we stop the optimization of model parameters.

## Data availability

Operational data of the HGF are publicly available through the California Department of Conservation Geologic Energy Management Division (CalGEM) repository (https://www.conservation.ca.gov/calgem/). The trajectories of all wells are obtained from digitization of scanned drilling and completion reports in the CalGEM repository. Monthly injection/extraction rates of each well are calculated based on the corresponding volumes and operation days. Data on operation pressures are not accessible from the repository. The relocated earthquake catalog is available from the Southern California Earthquake Data Center at https://scedc.caltech.edu/data/alt-2011-dd-hauksson-yang-shearer.html. Vertical displacement time series of leveling benchmarks from 1994 to 2017 and InSAR-observed displacement maps in two time periods: May 2006 to September 2010 and April 2015 to December 2017, are provided by Mariana Eneva[23]. The InSAR observations from June 2018 to August 2019 are measured with Sentinel-1 satellite SAR images, which are freely available through the Copernicus Open Access Hub (https://scihub.copernicus.eu/dhus/#/home).

## Code availability

Our numerical simulations are implemented with the software COMSOL Multiphysics (version 6.0); the calibrated reservoir model is available from Zenodo (https://zenodo.org/records/11609229).

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

## Acknowledgements
We appreciate Mariana Eneva for assistance in obtaining leveling observations from 1994 to 2017 and InSAR observations of two time periods: May 2006 to September 2010, March 2015 to December 2017. This paper benefits from U.S. Geological Survey internal reviews by Ole Kaven, Fred Pollitz and Shane Detweiler. G. Jiang is supported by the National Natural Science Foundation of China (grants 42072245 and 42130101) and the Fundamental Research Funds for the Central Universities. Any use of trade, firm or product names is for descriptive purposes only and does not imply endorsement by the U.S. government.

## Author contributions
G.J. designed the reservoir model, performed numerical simulations and slip modeling, and wrote the manuscript. A.J.B. designed and supervised experimental tests, processed geodetic data, and coauthored the manuscript. R.J.S. processed seismic data and contributed to manuscript preparation. K.Z.M. performed slip modeling and contributed to manuscript preparation. J.T. contributed to manuscript preparation. A.C.B. gathered and processed operational data, and performed initial analyses of geodetic observations.

## Competing interests
The authors declare no competing interests.
