## [Peer review File · Nature Communications]

Relatively stable pressure effects and time-increasing thermal contraction control Heber geothermal field deformationEditorial Note: Parts of this Peer Review File have been redacted as indicated to remove third-party material where no permission to publish could be obtained.

REVIEWER COMMENTS

Reviewer #1 (Remarks to the Author):

Jiang et al. hypothesize that Thermal contraction drives long-term deformation of the Heber geothermal field in southern California. To test this hypothesis, they employ a thermo-poro-mechanical model and investigate surface deformation, injection/production, and hydrogeological datasets.

The hypothesis is plausible, and the overall approach is suitable. However, I found several shortcomings in this manuscript that I detailed below.

1- The manuscript is poorly structured, and observations, results, and interpretations are mixed and confusing. I read the manuscript several times to understand what the authors tried to do and learn the paper's logic. Authors need to heavily restructure the manuscript. Also, in every figure, they need to introduce all symbols, signs, and colors.

2- Throughout the manuscript, errors and uncertainties are disregarded. Without knowledge of observation errors and model uncertainties, it is not possible to distinguish between competing mechanisms (e.g., thermal contraction, pore pressure, and poroelastic stresses) that drive surface deformation.

3- The implemented model is not adequate for studying geometrical sites. As the authors provided in Figure 1b, a geothermal site is a zone of heat anomaly and, thus, different from the surrounding area in terms of mechanical properties and heat content. As is shown in similar cases (e.g., Im and Avouac et al. 2021 Natue), the mechanical model needs to be set up to account for property contrast within the reservoir and surrounding rocks. The model setting used here implements basic layered rheology, which I believe is inappropriate; thus, results are unreliable. This might also be why modeled vertical deformation deviates from observed in most places, particularly following 2010.

4- I could not find a model misfit plot in the manuscript that shows how the model and observations agree. Figure 2 and supplemental figures that show modeled and observed vertical land motion suggest that agreement is not good, particularly following 2010. Need to provide maps of model misfit distributions for different episodes and discuss the possible mismatches and causes. Also, I am unsure why vertical deformation is normalized (e.g., figure 2). For simplicity, authors can show observed and modeled vertical deformation and misfits along two profiles (south-north and east-west) for different time steps.

Reviewer #2 (Remarks to the Author):

Review of the manuscript "Thermal contraction gradually dominates Heber geothermal field deformation", by Jiang et al.

This is an interesting study that shows the importance of thermal effects on the subsurface response to geothermal energy operations. The study focuses on the commonly overlooked thermal effects related to cooling around injection wells. Despite thermal effects usually receive little attention, the findings in this paper are known and the Discussion could benefit from putting in context the findings of this study with published work. In particular, the seismicity rate increase in the long term in geothermal systems has been shown to be due to cooling by Parisio et al. (2019); cooling effects on induced seismicity have also been shown at The Geysers (Jeanne et al., 2015); and the effect of cooling in the far field, i.e., in areas not affected by cooling, has the potential to reactivate distant faults in the long term (Kivi et al., 2022). Overall, I think this is an interesting study that could be accepted for publication provided that the following comments are addressed.

Major comments:

- Some relevant previous studies highlighting the importance of thermal effects in the long term are missing
- The Abstract should emphasize more the “long-term” effect of cooling
- Lines 145-150: it should be explained why there is subsidence from the beginning in the west region, where injection occurs and therefore uplift is expected. If subsidence is due to cooling, I would expect subsidence to be delayed for a few years until a relatively big volume of rock is cooled down and, thus, contracted
- A major limitation of the numerical simulations is the fact that fluid properties are not a function of pressure and temperature. According to the explanation of lines 282-288, fluid properties are constant. This assumption yields significant errors in terms of pore pressure distribution in regions with cooling because water viscosity increases at lower temperatures, affecting pore pressure gradients
- Lines 391-398: the reasons of the four observations should be explained

Minor comments:

- I recommend to avoid mixing units, e.g., °C and °F (better use °C in the whole manuscript)
- Figure 1d: the site has a tendency to subsidence, but this figure shows “normalized uplift”, which is confusing. It may be better to show “normalized subsidence” and also include the thermoelastic effect with negative values
- Line 231-232: please, state here which are the “three thermophysical parameters”
- Line 257 (and other lines in the manuscript): instead of “parameters”, the term “rock properties” could be used
- Typos: line 53: “refer it to” -> “refer to it”; line 113: “benchmarks, and has been” -> “benchmarks, has been”
- Caption Figure 4, b) I’d say that it should say “Temperature distribution at the same time” (without point)
- Caption Fig. 5: indicate where the location of the 8 probe points can be found

References

Jeanne, P., Rutqvist, J., Dobson, P. F., Garcia, J., Walters, M., Hartline, C. and Borgia, A. (2015). Geomechanical simulation of the stress tensor rotation caused by injection of cold water in a deep

geothermal reservoir. *Journal of Geophysical Research: Solid Earth*, 120(12), 8422-8438.

Kivi, I. R., Pujades, E., Rutqvist, J. and Vilarrasa, V. (2022). Cooling-induced reactivation of distant faults during long-term geothermal energy production in hot sedimentary aquifers. *Scientific Reports*, 12(1), 2065.

Parisio, F., Vilarrasa, V., Wang, W., Kolditz, O. and Nagel, T. (2019). The risks of long-term re-injection in supercritical geothermal systems. *Nature Communications*, 10(1), 4391.

Reviewer #3 (Remarks to the Author):

Please see attached pdf file.

[Editorial Note: This PDF is displayed across the next four pages]

Thermal contraction gradually dominates Heber geothermal field deformation

By G. Jiang et al.

General comments.

This manuscript presents a very interesting case study of geothermal long-term monitoring combining deformation, hydrological and seismicity data and their interpretation using thermo-hydro-mechanical modeling, in relation with the harnessing of the Heber reservoir and geothermal field in California, US. The presented data set is simply impressive by its time duration, demonstrating brilliantly that long term observations are required to understand the dynamics of a reservoir. The modelling performed seems apparently fine, although it is hard to judge without trying the codes. The number of tests performed is also impressive and the supplementary materials is an encyclopedia of tests in support of the results.

However, the manuscript suffers from issues on the content

- The title indicates temperature effect as the main cause. However, this is not clearly shown. The demonstration of the results is quite poor, possibly mainly because shown results are not properly explained. For example, at places it is hard to understand the link between the figure caption and what is shown on the figures. In addition, although we may guess it, it is not clear what is the general aim of the manuscript, except demonstrating a powerful method with just a case study. As a consequence, there is a lack of generalisation. What does this study bring for other geothermal fields? What are the lessons learned?
- There is often a mis-use of the word “deformation”, when referring to leveling and InSAR observations, which produce displacement data. This is general in the whole manuscript and also the Supplementary Materials. I have mentioned some of them, but please check thoroughly.
- I do not understand the ground reason of your area separations between west, central, east zones. This seems to be very artificial to me... The demonstration can be done without these artificial boundaries.
- You raise the point of global movements of the plate boundary, which is known to be pretty active, as your study addresses observations over several 10s years. I am not convinced that your displacement inversion can assess whether or not the Dixieland fault can be the reason of the observed changes, the cover of leveling network and InSAR coverage is simply much too small. Indeed, Dixieland fault takes place in a much larger regional networks of faults such as the Imperial and Weinert-El Centro faults, which seem to be at least as active than this one. To convince the reader, you need to refer to other studies and/or include additional data to exclude that there were no global movements during the several years. Actually, if there were, how would that affect your study? You need to reconsider your demonstration. In addition, in the supplementary material, your inversion procedure led me to conclude you may invert the wrong component (vertical), which you also corrected for signals possibly associated to what you want to invert for...
- Table S4 to S16. Impressive number of tests performed. However, it seems that some minima are not found as yet by lack of search in the model space. For example table S4.I. Looking at the column 16% for the upper reservoir, we see that the lower reservoir misfit decreases when the lower reservoir porosity decreases. May be if you would take 14% for the porosity of the lower reservoir, the misfit would be even smaller? How are you sure you found the minimal value of misfit for all parameters tested? This raises the question of the validity of the final result.

And the form

- There are many problems in the figures and their captions. Some figures are not clearly explained enough. Every single mention in the figure should be explained in the caption. Many are missing. Some figures are not showing what is said in the caption.
- When comparing figures of the same area for observations and interpolation or inversion, I would strongly suggest to use the same axis. Otherwise, it is challenging to find correspondence.

As a whole, this is an interesting case study, however the main message is not brought clearly enough, because the demonstration suffers from lack of clarity and some clear evidence that you reached the best model ultimately. In addition, whether the conclusion found can be generalized to other cases and how is not brought.

I suggest that the above mentioned issues and the specific comments below need to be solved before any decision can be taken on the manuscript.

Specific comments.

Line 49-52. The permeability is found along faults, which are described. However, it is not clear enough which is which with respect to the figure 1. In the text you do not mention the feeder fault, is it one on them?

Line 49-51. You describe the matrix permeability from 0.55 km until ~2 km (?) but you do not indicate what is above or below. This is lacking also here.

Line 51. "The normal fault" is very vague. Which one is it? How do you know it is a normal fault? What does it bring to the study?

Figure 1a. Please indicate what "F." means. In the caption, HGF meaning is missing. It should be given at the first occurrence in figures. HGF is missing in caption 1a. Same applies for SCSN: what is it? What are yellow circles? What are the beach balls? The plate boundary is certainly not so linear, is it? It would be more realistic to represent it with its real path, as feeder and subsidiary faults.

Figure 1d. The red line is usually called "cumulative seismicity" However, it is also often shown with the energy released. This information may not be available, as seismic events in geothermal system are small and estimating their energy sometimes challenging, but quantification would be nice here; it would be nice to indicate what is the variability of energies of the earthquakes shown. In the leveling time series, why the error bar increases with time and there is no error on the first point? The cyan color is hard to follow. May be plot it on top of the leveling points?

Line 70-71. I do not get it. The resource of a geothermal field cannot be designed... The resource is what we try to access, so we design the capacity of the power plant to match the resource potential. This sentence needs to be rephrasing. Or am I missing something here?

Line 72. There is a tendency now to try not to "exploit" natural resources, but rather harness. Could you check in all the manuscript?

Line 85-86. Possibly you need a reference here.

Line 86. It would be important to give information on the seismic network (number and type of stations) deployed since the beginning of the exploration and over the years. This would give us an

impression of if there is more earthquake due to the better recording capability of indeed because of more earthquakes.

Line 89. I would suggest to change “indicating” with “suggesting”.

Line 94-96. Actually in this paragraph, there is nothing new, this is known. Good to know for HGF, but so what?

Line 114, 139, 235, and other locations possibly, e.g., supplementary figure captions. A space missing between time and series.

Figure 2. The reason(s) of the separation of west, central and east regions is far for clear. From what is explained, there are no structural structures that justify this separation. Why not north, central and south?

Figure 2e. I do not understand why you need to normalize the vertical displacements. Note the term “vertical deformation” is inappropriate, as what is observed with both leveling and InSAR are displacements, and actually you indicate also “per year”, which makes it technically velocities.

Line 148. “The same scenario”. This is confusing. Which same scenario? In addition, if this is the same scenario between two areas separated the artificial boundaries (west and east), it means that there may be no need to separate them...

Figure 3. The figure shows a series of temperature points. However, there is not clarity... for instance in Fig. 3a, I do not see 11 injection wells values. Are there all mixed in the figure? Are they averaged? Or only one is shown? Similar issues for the other sub-figures.

Lines 169-171. These interesting observations are hard to believe, but is I generally trust data. This paragraph is just describing data, but do not suggest any potential reasons for the temperature changes. Possibly this is not the place, however, after displacement observations, you suggest a reason for the observed changes. Therefore I would expect the same for temperature observations. At line 178-180, there seems to be indeed a reason proposed: convection. Is that the case? If yes, I would make is even clearer.

Line 183. Indeed an active fault may move. It would be also interesting to relate the natural seismic activity along the entire Dixieland fault with time, to make the potential link between the local seismicity and the more regional one.

Line 186. Two major issues here:

1. I would be very cautious to invert local data for global interpretation.
2. Inversion of leveling data does not account for horizontal displacements. However, due to the local geodynamics setting strike-slip faults (e.g., the Dixieland fault?) mostly horizontal displacement would be expected.

Line 197. How did you measure horizontal displacements? The explanation resides in the InSAR/leveling integration. At places, this is not clear enough on the temporal evolution of the measurements and how you inverted each period of time. How to combine both of them is sometimes vague.

Line 201. Minor point: I would suggest you indicate the code used here also (although is it given in sup info)

Line 223. “strike-slip” it is not clear which fault is which mechanism. You could introduce them better in figure 1.

Line 228. Why inverting for layer parameter is not feasible? Does it not depend on which ones? You need more explanations here.

Figure 4. What are P1, P2, P8 etc. Please report extensively all elements within the figure caption.

Figure 5. This figure is very unclear, mainly because the sub-figure are too small, and axis are really tiny. In addition, it is not possible to understand what are the different curves of different colors.

Line 358. Figure 15 does not exist.

Line 491. The initials of Jean-Philippe Avouac are J. P.

Supplementary Materials:

Please check the use of “deformation”.

In § Fault slip inversion: I have an issue on what you want to invert for and the procedure. You remove the geothermal signal. Why and how do you do this? Then you subtract the pre-2005 trends. I really do not understand why. This may be the signal you want to invert for... You may need a reference here to make sure they are the whole area signal. After removal, what is left??

in the second step and (1), you use “corrected vertical-component data”. Here 2 things: 1. what are data corrected for? The previous trends? 2. why do you invert only the vertical data? Inversion procedure with Okada model is clearly possible with all components, so I do not understand why only vertical. In addition, as the plate boundary is a strike-slip process, there may be little vertical signal but large horizontal one. So this inversion results may be completely wrong, as you invert the wrong component in which you removed some potentially valid signal... You need to review here seriously the approach.

In the thermo-poro-electricity formulation, in the last paragraph you indicate primary effects and secondary effects. You could list them, and justify how to classify them as primary or secondary.

Figure caption of S1, S3. Check the use of “deformation” here.

Figure caption of S1. The normalization method for uplift is unclear. Why do you need to normalize? Is the normalized value inverted for??

Figure S2. In the caption, please add the reference for the kriging here too. In the figure, you should get the same axis limits for both the data and the interpolated info.

Figure S5. The sub figure are too small, making this figure useless. You also need to tell readers when... time is not indicated or clear enough.

Figure S6. There are several place where the term “excess” is used for vertical displacement. I am not convince with excess it is. I did not get where this excess is coming from. Need more explanation here and before possibly.

Figure S7. Where is A-33? Show it on the figures.

Figure S16, S17, S18. The number of sub figure is large. You could gain place by removing the axis for the inner figure, as they are the same for all. This would make the figure much easier to read. In addition, on the all 3 pages for each figure, I count 96 (4*8*3) points. Not clear where the 3 otehrs are...

Response to Reviewer #1

Jiang et al. hypothesize that Thermal contraction drives long-term deformation of the Heber geothermal field in southern California. To test this hypothesis, they employ a thermo-poro-mechanical model and investigate surface deformation, injection/production, and hydrogeological datasets. The hypothesis is plausible, and the overall approach is suitable. However, I found several shortcomings in this manuscript that I detailed below.

Ans: Thank you very much for your efficient work of reviewing our MS and also for your constructive comments.

As you know that deformation of geothermal fields can be caused by reservoir pressure depletion, poroelastic and thermoelastic contraction, but **the spatial and temporal evolution of their strengths is still in doubt**. One major reason for this knowledge gap is due to the rarity of long-term seismic and geodetic observations combined with the extreme difficulty of setting up a realistic 3D thermo-hydro-geological model with limited subsurface geological and geophysical data (like reservoir geometry, formation lithology and fault distribution) needed to simulate decades of geothermal operations.

In this study, we quantify spatiotemporal evolution of the relative strengths of the three physical mechanisms for the Heber Geothermal Field (HGF) based on a 3D thermo-hydro-geological model, which is calibrated by multiple datasets, including geodetic observations over 20 years, precise well trajectories and operational records of fluid rates and temperatures. Our most important findings are:

- 1. The effects of pressure fluctuation and poroelastic response keep relatively stable in the operation history of the HGF. In contrast, the strength of thermal contraction effect increases with time: after ~30 years of operations, the thermal effect is 2-3 times larger than the two pressure effects.**
- 2. It is common that fluid injection causes surface uplift due to pore pressure increases within the target reservoir; however, the ground in west and east regions of the HGF was subsiding during some time periods with net increases of injected fluid. We show that the deformation anomaly is related to the effect of the HGF center siphoning fluid from surrounding regions and ever-increasing effect of thermal contraction.**
- 3. The long-term trends of surface deformation and seismicity growth at the HGF are controlled by the thermal effect. In contrast, the two pressure effects only drive instantaneous changes.**

We have thoroughly revised the MS based on the comments from you and the other two reviewers to make these points as clear as possible. Following are our point-by-point response.

1-1 The manuscript is poorly structured, and observations, results, and interpretations are mixed and confusing. I read the manuscript several times to understand what the authors tried to do and learn the paper's logic. Authors need to heavily restructure the manuscript.

Ans: We are really sorry that the original MS was not structured with a straightforward logic. During this revision, we rewrite the main text into three sections: (1) Introduction (without subheading), (2) Results with 7 subheaded sections, and (3) Discussion, based on the formatting instructions of Nature Communications.

In the Introduction section, we begin with the background of our study, and then present the operation history of the HGF, and a brief summary of our major results and conclusions of our study.

For the Results section, we formulate it with the following logic: (1) presenting the observation results of surface displacement and seismicity with two sections; (2) presenting the results of fault slip modeling (to test whether the observed surface displacement is due to potential slip on the two strike-slip and one normal-slip faults); (3) presenting the results of Thermo-Hydro-Mechanical (THM) modeling (including model setup, model calibration, and model fit); (4) presenting the simulation results of reservoir pressure and temperature; (5) presenting the quantification results of spatio-temporal evolution of Thermo-Poro-Elastic effects; (6) presenting the physical mechanisms of the HGF deformation.

In the Discussion section, we compare our major findings with previous studies, and discuss their implications for the safety of geothermal operation and mitigation of geohazards.

1-2 Also, in every figure, they need to introduce all symbols, signs, and colors.

Ans: Thank you for your suggestion. For this revision, we revised the figures to match the revisions to the main text, and we have made sure the meaning of each symbol is introduced in the captions.

2 Throughout the manuscript, errors and uncertainties are disregarded. Without knowledge of observation errors and model uncertainties, it is not possible to distinguish between competing mechanisms (e.g., thermal contraction, pore pressure, and poroelastic stresses) that drive surface deformation.

Ans: In our study, we used leveling and InSAR observations of surface displacement as well as well temperature records to constrain our 3D THM model. The observation errors of leveling data range from 0.2 cm to 0.5 cm. The uncertainties of InSAR-based velocities are less than 2 mm/yr depending on the satellite and observation periods. In the figure below (Response Figure 1), you can see that the errors are much less the displacement magnitudes of the HGF, especially in the main deforming region where signals are nearly an order of magnitude larger than the uncertainty. More importantly, as we show in the main text, the deformation trends measured by leveling are consistent with the InSAR results, which confirms that the surface displacement observations are robust.

Response Figure 1: Uncertainty in InSAR based ground velocities compared to the size of the velocity estimate. Top: East rates. Bottom: Vertical rates. ENV=Envisat; SNT1a=Sentinel-1 early time period (2015/05-2017/12); SNT1b=Sentinel-1 later time period (2018/06-2019/08).

For the temperature records, we do not have any information on their observation errors because they come from a public repository provided by the local regulatory body. But, you can see the records of different wells in each cluster are consistent with each other (Response Figure 2).

Response Figure 2: Operation temperature changes of the Heber Geothermal Field.

Unfortunately, it is computationally impractical to estimate the uncertainties of a FEM model, especially of a 3D model, and we did not find any related references offering a way to roughly estimate these. Alternatively, to improve the robustness of our results, we use the observations of surface deformation and well temperatures to calibrate our 3D model. In addition, we also test the influences of different parameters. Our calibrated model can fit the spatiotemporal variations of observed surface displacement and well temperatures, which indicates that we can use the model to quantify the spatiotemporal evolution of thermo-poro-elastic effects.

3-1 The implemented model is not adequate for studying geometrical sites. As the authors provided in Figure 1b, a geothermal site is a zone of heat anomaly and, thus, different from the surrounding area in terms of mechanical properties and heat content. As is shown in similar cases (e.g., Im and Avouac et al. 2021 Natue), the mechanical model needs to be set up to account for property contrast within the reservoir and surrounding rocks. The model setting used here implements basic layered rheology, which I believe is inappropriate; thus, results are unreliable.

Ans: Yes, a geothermal site is a region of heat anomaly; but it does not indicate that the hydrogeological properties of the heat region are different from the surrounding rocks. The heat anomaly is mainly associated with the location of deep source of heat fluid. For the HGF, current knowledges on the lateral variations of rock properties are limited. To honor the natural complexity of the system while keeping the problem numerically tractable, we simplify the study region with five layers: caprock at the surface, two reservoir layers, a basal layer, and basement rock. Additionally, we embed the two NW-trending strike-slip faults and the NE-trending feeder fault in our model as individual formations with a unique set of thermo-hydro-mechanical properties. The NE-trending feeder fault is the primary pathway for deep, high-temperature brine upwelling into the shallow reservoir.

In our 3D model, the parameter settings mainly reflect the rock properties within the geothermal field rather than the surrounding region. The major reasons for such setting are: (1) obvious surface deformation only occurred within the field; (2) we only focus on the deformation mechanisms of the field; (3) only the deformation observations and temperature records within the field are used to calibrate our model; (4) we know little about the rock properties of the surrounding region.

3-2 This might also be why modeled vertical deformation deviates from observed in most places, particularly following 2010.

Ans: During our model calibration, we simulate the operation of the HGF from Apr 1985 through Dec 2010; the simulation results are compared with the vertical displacement timeseries (Jan 1994 - Nov 2010) at 64 leveling benchmarks, which are selected to give approximately uniform coverage within the HGF. The leveling observations after 2010 are not included for model calibration, and only used to validate our model.

We count the number of leveling stations with bad misfit of the model predictions to the observations. The total number is 30 (Figure S14). Most of these stations are located in the peripheral region of the HGF. In addition, there are 3 stations (1-A-5, 1-A-4, and C-4) located at the HGF center. The difference between the simulated and observed displacements in the peripheral regions is likely due to laterally inhomogeneous hydromechanical and thermophysical properties, and/or unmodeled variations in the

depth to basement rock of the study region. The effect of inhomogeneous rock properties increases with the duration of geothermal simulation. For the difference between the simulated and observed displacements at the HGF center, there is another reason that the leveling-observed subsidence is obviously larger than the InSAR monitoring results (Figures S2 and S3). Both of the two observation results are included during model calibration.

Overall, our model can replicate major features of observed surface displacement in both vertical and horizontal directions (Figs. 6 and S13).

4-1 I could not find a model misfit plot in the manuscript that shows how the model and observations agree. Figure 2 and supplemental figures that show modeled and observed vertical land motion suggest that agreement is not good, particularly following 2010. Need to provide maps of model misfit distributions for different episodes and discuss the possible mismatches and causes.

Ans: Thank you for your suggestion. We now plot a figure (Figure 6 in the main text) to show model predictions to surface deformation observations and model misfits.

4-2 Also, I am unsure why vertical deformation is normalized (e.g., figure 2).

Ans: The displacement time series of the leveling stations are normalized for two purposes primarily related to visualization: (1) to represent the deformation trends of the west, central and east zones of the HGF, and (2) to represent temporal variation of deformation of the whole region. By doing this, we can compare vertical deformation trends of the west, central and east regions with the respective net injection volumes of geothermal fluid of every month. In addition, we can also compare the deformation trend of the whole region with temporal changes of observed seismicity in the Heber geothermal field. To be clear, our misfit calculations are based on actual observations.

4-3 For simplicity, authors can show observed and modeled vertical deformation and misfits along two profiles (south-north and east-west) for different time steps.

Ans: Thank you for your suggestion. We plot the displacement profiles during this revision. Given that Figure 6 in the main text has shown the model predictions to surface deformation observation. We put the figure of displacement profiles in supplementary materias (Figure S15).

Response to Reviewer #2

This is an interesting study that shows the importance of thermal effects on the subsurface response to geothermal energy operations. The study focuses on the commonly overlooked thermal effects related to cooling around injection wells. Despite thermal effects usually receive little attention, the findings in this paper are known and the Discussion could benefit from putting in context the findings of this study with published work. In particular, the seismicity rate increase in the long term in geothermal systems has been shown to be due to cooling by Parisio et al. (2019); cooling effects on induced seismicity have also been shown at The Geysers (Jeanne et al., 2015); and the effect of cooling in the far field, i.e., in areas not affected by cooling, has the potential to reactivate distant faults in the long term (Kivi et al., 2022). Overall, I think this is an interesting study that could be accepted for publication provided that the following comments are addressed.

Ans: Thank you very much for your careful review and constructive comments.

During this revision, we read through the three papers you recommended and other literatures like Simone et al. (2013; <http://dx.doi.org/10.1016/j.pce.2013.01.001>). The major findings of these studies are summarized below:

- (1) Simone et al. (2013): thermal effects can induce a significant perturbation on the stress in the intact rock affected by the temperature drop; this perturbation is likely to trigger induced seismicity in the surroundings of critically oriented fractures near the injection well.
- (2) Jeanne et al. (2015): thermal effects can cause stress tensor orientation changes; the vertical stress reduction is significant and propagates far below the injection well.
- (3) Parisio et al. (2019): thermal effects dominate the geomechanical response of geothermal reservoirs.
- (4) Kivi et al. (2022): thermal stresses are transmitted much ahead of the cooled region and are likely to destabilize faults located far away from the reinjection well.

In contrast, **our study focuses on not only the thermal effect but also the pressure effects of pressure fluctuation and poroelastic response.** We try to resolve the physical mechanisms of observed surface displacement, especially of two abnormal phenomena: in the west and east regions of the Heber Geothermal Field, where only injection operations occur, the ground was subsiding from 1994 to 2005 and after 2005, respectively. In addition, we also strive to quantify the spatiotemporal evolution of relative strengths of three physical mechanisms: reservoir pressure depletion, poroelastic effects, and thermoelastic contraction.

Although we now know that deformation of geothermal fields can be caused by the pressure and thermal effects, **the spatial and temporal evolution of their strengths has not been well understood.** One major reason for this knowledge gap is due to the rarity of long-term geodetic observations combined with the extreme difficulty of setting up a realistic 3D thermos-hydro-geological model with limited subsurface geological and geophysical data (like reservoir geometry, formation lithology and fault distribution) needed to simulate decades of geothermal operations. With exception to Jeanne et al. (2015), the published studies above were conducted with 2D conceptual models. While they represent important advances for the reasons mentioned above, we feel that they are not sufficient to characterize the spatiotemporal variation of different physical mechanisms. In the discussion section of the revised manuscript, we now cite the papers on the thermal effect and compared our results with their findings.

Following are our point-by-point responses to the more specific comments:

Major comments:

1. Some relevant previous studies highlighting the importance of thermal effects in the long term are missing

Ans: Thank you for your suggestion. We cited the following papers on the thermal effect:

(1) De Simone, S., Vilarrasa, V., Carrera, J., Alcolea, A. & Meier, P. Thermal coupling may control mechanical stability of geothermal reservoirs during cold water injection. *Phys. Chem. Earth* 64, 117–126 (2013).

(2) Jeanne, P., Rutqvist, J., Dobson, P. F., Garcia, J., Walters, M., Hartline, C. and Borgia, A. (2015). Geomechanical simulation of the stress tensor rotation caused by injection of cold water in a deep geothermal reservoir. *Journal of Geophysical Research: Solid Earth*, 120(12), 8422-8438.

(3) Kivi, I. R., Pujades, E., Rutqvist, J. and Vilarrasa, V. (2022). Cooling-induced reactivation of distant faults during long-term geothermal energy production in hot sedimentary aquifers. *Scientific Reports*, 12(1), 2065.

(4) Parisio, F., Vilarrasa, V., Wang, W., Kolditz, O. and Nagel, T. (2019). The risks of long-term re-injection in supercritical geothermal systems. *Nature Communications*, 10(1), 4391.

We think this highlights the importance of thermal effects, as suggested, and also helps clarify the novelty of our study.

2. The Abstract should emphasize more the “long-term” effect of cooling

Ans: Thank you for your suggestion. In the revised abstract, which has a limit no more than 150 words, we emphasized the effects from the following two aspects:

(1) thermal contraction increases gradually with time and eventually overwhelms the pressure effects of pressure fluctuation and poroelastic response, which keep relatively stable during geothermal operations;

(2) thermal contraction dominates long-term trends of surface displacement and seismicity growth, while the pressure effects drive near-instantaneous changes.

3. Lines 145-150: it should be explained why there is subsidence from the beginning in the west region, where injection occurs and therefore uplift is expected. If subsidence is due to cooling, I would expect subsidence to be delayed for a few years until a relatively big volume of rock is cooled down and, thus, contracted

Ans: The reason why ground in the west region was subsiding from 1994 to 2005 with a net increase of injected fluid is that the average injection rate (0.6×10^9 kg/month) in the west region is much less than the production rates in the central region ($\sim 3.5 \times 10^9$ kg/month). The high production rates at the HGF center caused a pressure drop of up to 3 MPa relative to the initial pressure of 14.7 MPa (Response Figure 3); this created a low-pressure center that siphons fluid from surrounding regions. The fluid volume siphoned by the HGF center from the west region are more than the injection volume before 2005.

Response Figure 3: Simulated pressure changes at a time node during 1994 and 2005.

4. A major limitation of the numerical simulations is the fact that fluid properties are not a function of pressure and temperature. According to the explanation of lines 282-288, fluid properties are constant. This assumption yields significant errors in terms of pore pressure distribution in regions with cooling because water viscosity increases at lower temperatures, affecting pore pressure gradients

Ans: During this revision, we discussed this problem with Josh Taron from USGS, and add him to be our coauthor. As the functions of fluid properties changing with pressure and temperature are not available, we instead test the influences of different values of water properties on simulation results of surface displacement and reservoir temperature. From table S16, you can see that the influence is little.

Table S16. Misfits of model predictions with different properties of water.

RMS misfit	Density (kg/m ³)			Viscosity (Pa·s)				Compressibility (Pa ⁻¹)		
	800	900	1000	1.5×10 ⁻⁴	2×10 ⁻⁴	2.5×10 ⁻⁴	3×10 ⁻⁴	3.7×10 ⁻¹⁰	4.2×10 ⁻¹⁰	4.7×10 ⁻¹⁰
Leveling (cm)	2.83	2.93	2.39	2.39	2.39	2.39	2.39	2.39	2.39	2.39
InSAR (cm)	2.90	3.13	2.68	2.68	2.68	2.68	2.68	2.68	2.68	2.68
Weighted Misfit (cm)	3.56	3.71	3.06	3.06	3.06	3.06	3.06	3.06	3.06	3.06
Temperature (°C)	20.77	20.71	20.30	20.30	20.30	20.30	20.30	20.30	20.30	20.30
RMS misfit	Thermal Conductivity (W/(m·°C))			Specific Heat (J/(kg·°C))						
	0.58	0.68	0.78	3900	4200	4500	4800			
Leveling (cm)	2.39	2.39	2.39	2.63	2.39	2.69	2.81			
InSAR (cm)	2.68	2.68	2.68	2.57	2.68	2.97	2.55			
Weighted Misfit (cm)	3.06	3.06	3.06	3.27	3.06	3.43	3.45			
Temperature (°C)	20.30	20.30	20.30	20.49	20.30	20.56	20.88			

P.S. The parameter values in our tests are set based on the reservoir temperature (injection: 75-85°C; production: 150-175°C) and pressure (~12-18 MPa).

5. Lines 391-398: the reasons of the four observations should be explained

Ans: Thank you for your suggestion. The reasons were presented in the last section of the original MS (*Spatiotemporal evolution of thermo-poro-elastic effects*). We would like to illustrate the reasons for the interesting phenomena here.

- (1) The first phenomenon that ground was subsiding during some time periods with pressure increase, especially at points P5 and P6, is due to the effect of thermal contraction.
- (2) The two phenomena that surface subsidence at point P3 with ~ 1.3 MPa of pressure decrease and 1.6°C of temperature decrease is comparable to the subsidence at point P7 with ~ 1.8 MPa of pressure increase and 53.5°C of temperature decrease, and that surface subsidence at point P6 is up to 34 cm with only 3°C of temperature decrease and ~ 1.5 MPa of pressure increase, are also caused by the cooling effect, whose strength depends on temperature decrease and varies with locations.
- (3) The last phenomenon that temperature decrease at point P5 started from 2009, ~ 13 years later than the time of ground subsidence occurred, is caused by the fact that the thermal contraction effect can disturb broad ranges over the temperature decrease front.

Response Figure 4: Temporal evolution of pressure, temperature and their effects at four probe points.

Our original intention of introducing these phenomena is to show the features of the thermal effect:

- (1) thermoelastic effect varies nonlinearly and asynchronously with temperature;
 - (2) thermal contraction effect can disturb broad ranges over the temperature decrease front.
- These two features cause prominent thermal effect at some locations with little temperature decrease.

During this revision, we realized that the original description is hard for readers to follow, and rewrote the paragraph. You can find it in the last paragraph of the section “*Quantification of inhomogeneous Thermo-Poro-Elastic effects*”.

Minor comments:

1. I recommend to avoid mixing units, e.g., $^\circ\text{C}$ and $^\circ\text{F}$ (better use $^\circ\text{C}$ in the whole manuscript)

Ans: Thank you for your suggestion. We made units consistent, replacing the unit of °F with °C where necessary.

2. Figure 1d: the site has a tendency to subsidence, but this figure shows “normalized uplift”, which is confusing. It may be better to show “normalized subsidence” and also include the thermoelastic effect with negative values

Ans: Thank you for your suggestion. The label “normalized uplift” is replaced with “normalized U_z ”. The vertical displacements (U_z) associated with the thermoelastic effect are shown with negative values in the new Figure 4c.

3. Line 231-232: please, state here which are the “three thermophysical parameters”

Ans: Thank you for your suggestion. The three thermophysical parameters are specific heat, thermal conductivity and thermal expansion coefficient. We added them into the corresponding sentence in the revised MS.

4. Line 257 (and other lines in the manuscript): instead of “parameters”, the term “rock properties” could be used

Ans: Thank you for your suggestion. The word “parameters” was replaced with “rock and water properties” in the revised MS. In addition, we eliminated the use of ambiguous words.

5. Typos: line 53: “refer it to” -> “refer to it”; line 113: “benchmarks, and has been” -> “benchmarks, has been”

Ans: Thank you for your careful review. We corrected these errors during this revision.

6. Caption Figure 4, b) I'd say that it should say “Temperature distribution at the same time” (without point)

Ans: Thank you for your suggestion. We revised the sentence in the figure caption.

7. Caption Fig. 5: indicate where the location of the 8 probe points can be found

Ans: Thank you for your suggestion. During this revision, we replaced the original Figure 5 with new Figures 7 and 8, where we plot the location of the 8 probe points.

Response to Reviewer #3

This manuscript presents a very interesting case study of geothermal long-term monitoring combining deformation, hydrological and seismicity data and their interpretation using thermo-hydro-mechanical modeling, in relation with the harnessing of the Heber reservoir and geothermal field in California, US. The presented data set is simply impressive by its time duration, demonstrating brilliantly that long-term observations are required to understand the dynamics of a reservoir. The modelling performed seems apparently fine, although it is hard to judge without trying the codes. The number of tests performed is also impressive and the supplementary materials are an encyclopedia of tests in support of the results.

Ans: We appreciate your careful review and insightful comments, which indeed help improve the quality of our MS. We also appreciate the recognition that long-term observations and sophisticated numerical models are needed to understand geothermal hazards. For some additional context:

We started this project in 2017, and spent great efforts to collect and analyze Leveling and InSAR observations of more than 20 years. We found that the west, central and east regions of the Heber Geothermal Field (HGF) experienced different deformation trends. In doing so we uncovered a striking and previously unrecognized phenomenon of deformation trend reversals. Further, we found that the west and east regions were subsiding during the time periods with a net increase of injected fluid. In addition, we found that the growth of observed seismicity is consistent with surface displacement trend through normalization analysis of leveling time-series.

It is also time-consuming and challenging work to set up a realistic 3D hydrogeological model for the HGF and to calibrate its parameters with multiple datasets including surface deformation observations, precise well trajectories and operational records of fluid rates and temperatures. The main purposes of our study are: (1) to quantify the spatiotemporal evolution of relative strengths of three physical mechanisms: reservoir pressure depletion, poroelastic effects, and thermoelastic contraction; (2) and to resolve the physical mechanisms of complex surface displacement trends observed in different locations and time periods.

Our numerical simulations are implemented with the software COMSOL Multiphysics (version 6.0). The model files are available from Zenodo (<https://zenodo.org/records/10460410>). We revised the MS based on the constructive and comprehensive comments from you and the other two reviewers. Following are our point-by-point responses.

However, the manuscript suffers from issues on the content

1-1. The title indicates temperature effect as the main cause. However, this is not clearly shown. The demonstration of the results is quite poor, possibly mainly because shown results are not properly explained.

Ans: Thank you for this point. The meaning of the original title is that the strength of thermal effect increases gradually with time and eventually becomes the dominant cause. But, to avoid misleading the readers, we revised the title into “*Relatively stable pressure effects and time-increasing thermal contraction control Heber geothermal field deformation*”, based on our major findings:

1. The effects of pressure fluctuation and poroelastic response keep relatively stable in the operation history of the HGF. In contrast, the strength of thermal contraction effect increases

with time: after ~30 years of operations, the thermal effect is 2-3 times larger than the two pressure effects.

2. The long-term trends of surface deformation and seismicity growth at the HGF are controlled by the thermal effect. In contrast, the two pressure effects only drive instantaneous changes.
3. It is common that fluid injection causes surface uplift due to pore pressure increases within the target reservoir; however, the ground in west and east regions of the HGF was subsiding during some time periods with net increases of injected fluid. Here we show that the deformation anomaly is related to the effect of the HGF center siphoning fluid from surrounding regions and ever-increasing effect of thermal contraction.

We apologize that the original MS was not structured with a straightforward logic. For this revision we significantly overhauled the structure of the text and revised the figures. We re-wrote the main text into three sections: (1) Introduction (without subheading); (2) Results with 7 subheaded sections; (3) Discussion, based on the formatting instructions of Nature Communications.

In the Introduction section, we begin with the background of our study, and then present the operation history of the HGF, and a brief summary of our major results and conclusions of our study.

For the Results section, we formulate it with the following logic: (1) presenting the observation results of surface displacement and seismicity with two sections; (2) presenting the results of fault slip modeling (to test whether the observed surface displacement is due to potential slip on the two strike-slip and one normal-slip faults); (3) presenting the results of Thermo-Hydro-Mechanical (THM) modeling (including model setup, model calibration, and model fit); (4) presenting the simulation results of reservoir pressure and temperature; (5) presenting the quantification results of spatio-temporal evolution of Thermo-Poro-Elastic effects; (6) presenting the physical mechanisms of the HGF deformation.

In the Discussion section, we compare our major findings with previous studies, and discuss their implications for the safety of geothermal operation and mitigation of geohazards.

1-2. For example, at places it is hard to understand the link between the figure caption and what is shown on the figures.

Ans: Again, we apologize that our original figures were not presented clearly. For this revision, we replotted the figures to support/illustrate the major points of the main text. The captions were also rewritten to give clear explanations to each symbol in the figures.

1-3. In addition, although we may guess it, it is not clear what is the general aim of the manuscript, except demonstrating a powerful method with just a case study. As a consequence, there is a lack of generalisation. What does this study bring for other geothermal fields? What are the lessons learned?

Ans: Thank you very much for these points. It is clear to us now that the manuscript needed to be restructured to make the motivation clear. Our study has two major aims: (1) to quantify the spatiotemporal evolution of relative strengths of three physical mechanisms: reservoir pressure depletion, poroelastic effects, and thermoelastic contraction; (2) and to resolve the physical mechanisms of complex surface deformation trends observed in different locations and time periods.

With the revised manuscript, we improved the generalisation of our MS by comparing our major findings with previous studies. In addition, we add Discussion section to introduce the implications of our method and major findings. To reiterate:

- (1) As you know, although the pressure and thermal effects have been recognized as the causal mechanisms of geothermal field deformation, the spatial and temporal evolution of their strengths was not well understood. One major reason for this knowledge gap is due to the rarity of long-term seismic and geodetic observations combined with the extreme difficulty of setting up a realistic 3D thermo-hydro-geological model with limited subsurface geological and geophysical data (like reservoir geometry, formation lithology and fault distribution) needed to simulate decades of geothermal operations. Our framework developed for resolving the spatiotemporal evolution of the HGF deformation mechanisms through integrating multiple geodetic, geophysical and geological data is applicable for other complex geotherm systems.
- (2) Benefited from the 3D realistic THM model of the HGF, we find distinct features of pressure and thermal effects. First, the pressure effects respond much faster to changes in geothermal operations than the thermal effect. Second, different from the instantaneous response and proportional relation of poroelastic effects to pressure fluctuation, the effect of thermal contraction varies nonlinearly and asynchronously with temperature. Third, the pressure effects keep relatively stable during geothermal operations whereas the thermal effect increases gradually with time and eventually overwhelms the pressure effects. Lastly, thermal contraction dominates long-term growth of seismicity, while the pressure effects drive near-instantaneous changes.

These major findings advance current understanding of deformation mechanisms associated with geothermal energy production. In general the pressure effects are dominant at the early stage of geothermal operations, whereas the thermal effect increases with time and eventually dominates. In addition, these findings also provide theoretical implications for mitigating geohazards at geothermal fields and for improving the sustainability and safety of geothermal operations.

2. There is often a mis-use of the word “deformation”, when referring to leveling and InSAR observations, which produce displacement data. This is general in the whole manuscript and also the Supplementary Materials. I have mentioned some of them, but please check thoroughly.

Ans: Thank you for your careful review. We agree, and where appropriate we replaced “deformation” with “displacement” when describing the leveling and InSAR observation results.

3. I do not understand the ground reason of your area separations between west, central, east zones. This seems to be very artificial to me... The demonstration can be done without these artificial boundaries.

Ans: We divide the study area into the three zones based on two main reasons: (1) the west, central and east regions of the Heber geothermal fields experienced different displacement trends (for more details please refer to our response to your general comments); and (2) there are differences in geothermal operations across the field, with only injection operations in the west and east zones, but both injection and extration operations occurred in the central zones.

4. You rise the point of global movements of the plate boundary, which is known to be pretty active, as your study addresses observations over several 10s years. I am not convinced that the your displacement

inversion can assess whether or not the Dixieland fault can be the reason of the observed changes, the cover of leveling network and InSAR coverage is simply much too small. Indeed, Dixieland fault takes place in a much larger regional networks of faults such as the Imperial and Weinert-El Centro faults, which seem to be at least as active than this one. To convince the reader, you need to refer to other studies and/or include additional data to exclude that there were no global movements during the several years. Actually, if there were, how would that affect your study? You need to reconsider your demonstration. In addition, in the supplementary material, your inversion procedure lead me to conclude you may invert the wrong component (vertical), which you also corrected for signals possibly associated to what you want to invert for...

Ans: Thank you for these important comments. In our study region the leveling benchmarks are distributed within a region of $\sim 10\text{km} \times 10\text{km}$, but the coverage area of InSAR data is much larger, over $\sim 20\text{km} \times 20\text{km}$. These observations, especially InSAR, have good coverage of the southeast extension of the plate boundary (Dixieland) fault. The surface displacements associated with potential fault slip should be distributed along the faults. However, we only observed surface displacements at the HGF, which are predominately subsidence and uplift. In the other regions, we did not observe any deformation signals, which can represent the “global movements”. In other words, the observed geodetic anomalies occurred within a localized zone within the HGF on a far-smaller scale than the Dixieland fault.

To convince the reader, we further conduct three scenarios of fault slip modeling to available displacement observations. First, the long-term vertical displacement measured by leveling from 1994 to 2004 is inverted with normal slip on the feeder fault. Second, the vertical transient displacement from 2006 to 2010, obtained through removing the long-term displacement trends, is inverted with normal slip on the feeder fault. Third, the horizontal displacement measured by InSAR from 2006 to 2010 is modelled with the strike-slip Dixieland fault. These modeling can prove whether the long-term and transient displacement is associated with fault slip.

Our inversions show that neither the long-term vertical displacement nor the transient displacement observation can be reasonably fit by normal slip on the feeder fault. For the InSAR-measured horizontal displacements, the magnitudes of dextral strike-slip on the plate boundary fault need to be up to 30 to 82 cm below the depth of 8 km, which suggests 6 to 16 cm/yr of slip rates, much higher than the reported slip rates and the relative plate motion rate (only ~ 4 cm/yr).

5. Table S4 to S16. Impressive number of tests performed. However, it seems that some minima are not found as yet by lack of search in the model space. For example table S4.I. Looking at the column 16% for the upper reservoir, we see that the lower reservoir misfit decreases when the the lower reservoir porosity decreases. May be if you would take 14% for the porosity of the lower reservoir, the misfit would be even smaller? How are you sure you found the minimal value of misfit for all parameter tested? This rises the question of the validity of the final result.

Ans: We understand your concern. During this revision, we conduct more simulations with smaller porosities of the two reservoir layers during the first round of model calibration (see the following table). You can see that the RMS misfit to surface displacement observations is 4.01 cm with the porosity of the lower reservoir equal to 14%

Table S4. Misfits of model predictions with different reservoir porosities to displacement and temperature observations. Orange and green values correspond to the first and second rounds of model calibration, respectively.

I. RMS misfits to Leveling observations (cm)					II. RMS misfits to InSAR observations (cm)						
Porosity	Upper reservoir				Porosity	Upper reservoir					
	14%	16%	18%	20%		14%	16%	18%	20%		
Lower reservoir	14%	3.17	3.33	3.13	3.38	Lower reservoir	14%	2.62	2.71	2.38	2.54
	16%	3.30	2.98	3.21	3.27		16%	2.48	2.69	2.69	2.62
	18%	3.72	3.10	2.90	3.43		18%	3.47	2.68	2.86	2.52
	20%	3.51	3.44	3.02	3.56		20%	2.87	3.04	2.68	2.63
		3.18	2.80	2.80			2.63	2.81	2.47		

III. Weighted RMS misfits of deformation (cm)					IV. RMS misfits to well temperature (°C)						
Porosity	Upper reservoir				Porosity	Upper reservoir					
	14%	16%	18%	20%		14%	16%	18%	20%		
Lower reservoir	14%	3.83	4.01	3.73	4.02	Lower reservoir	14%	21.25	22.51	21.67	21.15
	16%	3.92	3.65	3.88	3.92		16%	21.19	21.15	21.48	20.79
	18%	4.59	3.77	3.61	4.06		18%	20.95	21.16	21.55	20.71
	20%	4.23	4.20	3.69	4.22		20%	21.40	21.59	20.96	20.81
		3.84	3.50	3.42			20.57	20.26	20.90		

We realized that there would be some potential uncertainties of the optimal solution associated with the choose of test parameter values. We thus conduct two rounds of model calibration to minimize potential uncertainties.

And the form

- There are many problems in the figures and their captions. Some figures are not clearly explained enough. Every single mention in the figure should be explained in the caption. Many are missing. Some figures are not showing what is said in the caption.

Ans: During the revision, we replotted the figures in the main text and rewrote the corresponding captions.

- When comparing figures of the same area for observations and interpolation or inversion, I would strongly suggest to use the same axis. Otherwise, it is challenging to find correspondence.

Ans: Thank you for your suggestion. During the process of reploting the displacement maps, we used the same spatial range.

As a whole, this is an interesting case study, however the main message is not brought clearly enough, because the demonstration suffers from lack of clarity and some clear evidence that you reached the best model ultimately. In addition, whether the conclusion found can be generalized to other cases and how is not brought. I suggest that the above mentioned issues and the specific comments below need to be solved before any decision can be taken on the manuscript.

Ans: We are very appreciative for your careful review. As we mentioned before, the MS has been revised based on the constructive and comprehensive comments from you and the other two reviewers to address these issues of clarity and supporting evidence. We believe the revised manuscript allows the readers to clearly understand the motivation, the data, the method, and the main results with clear supporting evidence and robust conclusions.

Specific comments.

Line 49-52. The permeability is found along faults, which are described. However, it is not clear enough which is which with respect to the figure 1. In the text you do not mention the feeder fault, is it one on them?

Ans: The feeder fault is the normal-slip fault. We introduced it in Line 53 of the original MS. During this revision, we add the symbols for strike-slip faults and the normal-slip feeder fault into Figure 1a.

Line 49-51. You describe the matrix permeability from 0.55 km until ~2 km (?) but you do not indicate what is above or below. This is lacking also here.

Ans: The matrix permeability structure consists of a shallow sedimentary reservoir and three faults. Above and below the reservoir are capping clays and basement, respectively (Response Figure 5).

[REDACTED]

Response Figure 5: Temporal evolution of pressure, temperature and their effects at four probe points. Simplified geologic and hydrothermal profile of the HGF (modified from James et al., 1987).

Line 51. “The normal fault” is very vague. Which one is it? How do you know it is a normal fault? What does it bring to the study?

Ans: The normal fault is the feeder fault. We label it in Figure 1a. James et al. (1987) presented this fault in their figure 1.

[REDACTED]

Response Figure 6

Figure 1a. Please indicate what “F.” means. In the caption, HGF meaning is missing. It should be given at the first occurrence in figures. HGF is missing in caption 1a. Same applies for SCSN: what is it? What are yellow circles? What are the beach balls? The plate boundary is certainly not so linear, is it? It would be more realistic to represent it with its real path, as feeder and subsidiary faults.

Ans: Thank you for your careful review. “F.” indicates “Fault”. HGF is Heber geothermal field; we defined it in the main text. SCSN is Southern California Earthquake Data Center; we add the full name into the caption of Figure 1a. The yellow circles are SCSN catalog. Beach balls are focal mechanisms of five $M > 2$ earthquakes. For the plate boundary (Dixieland) fault, there is no relative information on its surface trace in our study region. So we simplify it as a line based on its northwest extension and Figure 1 in James et al. (1987).

Figure 1d. The red line is usually called “cumulative seismicity” However, it is also often shown with the energy released. This information may not be available, as seismic events in geothermal system are small and estimating their energy sometimes challenging, but quantification would be nice here; it would be nice to indicate what is the variability of energies of the earthquakes shown. In the leveling time series, why the error bar increases with time and there is no error on the first point? The cyan color is hard to follow. May be plot it on top of the leveling points?

Ans: Thank you for your suggestion. We revised the label of left axis into “cumulative seismicity”. We agree that having some estimates of seismic energy release would be interesting, but as you mentioned this topic is deserving of its own study, especially for geothermal fields. We are planning to write a follow up study focusing on the induced seismicity, and we prefer to leave the energy calculations in that study to keep the scope of the present this study tractable to understand.

The “error bar” are derived from the fluctuation of leveling time series (see Response Figure 7). From the following figure, you can see that, for the first point, the displacement of each benchmark is equal

to 0. So the “error” is 0. The “error bar” only reflect the fluctuation of leveling time series rather than the observation error. The observation errors of leveling data range from 0.2 cm to 0.5 cm.

Response Figure 7: Spatial distribution of leveling stations and time series of observed vertical displacement.

The cyan color is changed to blue color. We plot the curve on top of the leveling points.

Line 70-71. I do not get it. The resource of a geothermal field cannot be designed... The resource is what we try to access, so we design the capacity of the power plant to match the resource potential. This sentence needs to be rephrasing. Or am I missing something here?

Ans: Thank you for your careful review. We revised the sentence to be “*The HGF resource was initially estimated to support a capacity of up to 500 MW of electrical power generation.*”

Line 72. There is a tendency now to try not to “exploit” natural resources, but rather harness. Could you check in all the manuscript?

Ans: Thank you for your suggestion. We deleted the word “exploit” in the revised MS. We also checked the MS and confirm that the word “exploit” is not used in the other places.

Line 85-86. Possibly you need a reference here.

Ans: Thank you for your suggestion. We rewrote the paragraph during the revision.

Line 86. It would be important to give information on the seismic network (number and type of stations) deployed since the beginning of the exploration and over the years. This would give us an impression of if there is more earthquake due to the better recording capability of indeed because of more earthquakes.

Ans: This is a good point. The stations used are from the Southern California Seismic Network (SCSN; see <https://www.fdsn.org/networks/detail/CI/>), and for this revision, we compared the network catalog with a matched-filter catalog based on cross-correlation of template events with continuous data. The methodology, which we now describe in the Methods section, adjusts for network coverage as it scans through continuous data. The two catalogs show a similarly increasing trend of seismicity, although the MF catalog provides better temporal resolution, which confirms the trends in seismicity are real rather than artifacts of network coverage. In addition, the template matched catalog shows that rapid increase of earthquake within the template matched catalog occurred at the time with prominent changes of injection and production rates.

Line 89. I would suggest to change “indicating” with “suggesting”.

Ans: OK. Thank you!

Line 94-96. Actually in this paragraph, there is nothing new, this is known. Good to know for HGF, but so what?

Ans: During this revision, we deleted the paragraph and put the relative contents on surface displacement and seismicity into the first two subsections of “Results”, respectively.

From the observations of surface displacement and seismicity, we found an interesting phenomenon that the temporal growth of seismicity rates is similar to the overall trend of ground deformation shown by the normalized leveling time-series, suggesting that physical mechanisms of surface deformation also control the occurrence of earthquakes at the HGF.

Line 114, 139, 235, and other locations possibly, e.g., supplementary figure captions. A space missing between time and series.

Ans: Thank you for your careful review! We changed the word “timeseries” into “time series” in the revised MS.

Figure 2. The reason(s) of the separation of west, central and east regions is far for clear. From what is explained, there are no structural structures that justify this separation. Why not north, central and south?

Ans: As we introduce previously, the study area is divided into the three zones based on two main reasons: (1) the west, central and east regions of the Heber geothermal fields experienced different displacement trends, and (2) there are only injection operations in the west and east zones, but both injection and extration operations occurred in the central zones.

Figure 2e. I do not understand why you need to normalize the vertical displacements. Note the term “vertical deformation” is inappropriate, as what is observed with both leveling and InSAR are displacements, and actually you indicate also “per year”, which makes it technically velocities.

Ans: The displacement time series of the leveling stations are normalized for two purposes primarily related to visualization of a complex history of displacements in space and time: (1) we needed a way to represent the displacement trends of the west, central and east regions of the Heber geothermal field, which show different displacement trends; and (2) to represent temporal variation of displacement of the whole region. By doing this we can: (1) compare vertical displacement trends of the west, central and east regions with the respective net injection volumes of geothermal fluid of every month; and (2) compare the displacement trend of the whole region with temoporal changes of observed seismicity at the Heber geothermal field. Again, we stress that our misfit calculations are based on the true displacements rather than the normalizaed ones.

In addition, to compare the observation results of leveling and InSAR, the InSAR LOS displacement rate maps of ascending and descending tracks are decomposed into vertical and east components. The leveling-measured displacement rate maps are calculated based on the cumulative displacements of benchmarks and the corresponding observation durations.

Based on your suggestions, we replaced the word “vertical deformation” is replaced with “vertical displacement”.

Line 148. “The same scenario”. This is confusing. Which same scenario? In addition, if this is the same scenario between two areas separated the artificial boundaries (west and east), it means that there may be no need to separate them...

Ans: “The same scenario” indicates that ground was susiding with fluid injection. We revised the words into “The same phenomenon”. Although this phenomenon has been observed in both west and east zones, it occurred in different time periods: 1994-2005 (west) and 2005 to now (east). More importantly, the deformation anomalies are caused by different mechansims: the siphonic effect of the HGF center (west) and thermoelastic contraction (east).

The reasons for dividing the study area into 3 zones refer to our above responses to your comment on Figure 2e.

Figure 3. The figure shows a series of temperature points. However, there is not clarity... for instance in Fig. 3a, I do not see 11 injection wells values. Are there all mixed in the figure? Are they averaged? Or only one is shown? Similar issues for the other sub-figures.

Ans: Thank you for your comments. The temperature observations of injection and extraction wells are mixed in Figure 3. We clarify it in the caption.

Lines 169-171. These interesting observations are hard to believe, but is I generally trust data. This paragraph is just describing data, but do not suggest any potential reasons for the temperature changes. Possibly this is not the place, however, after displacement observations, you suggest a reason for the observed changes. Therefore I would expect the same for temperature observations.

Ans: In our assessment, the temperature changes of the 11 deviated wells are probably associated with geothermal operations. Consequently, for this revision, we put the contents on temperature records into the section of Introduction as a background of the HGF.

At line 178-180, there seems to be indeed a reason proposed: convection. Is that the case? If yes, I would make is even clearer.

Ans: Yes, you are right. During this revision, we explain the reason for the temperature changes in the subsection of Results (*Reservoir pressure and temperature evolution*).

Line 183. Indeed an active fault may move. It would be also interesting to relate the natural seismic activity along the entire Dixieland fault with time, to make the potential link between the local seismicity and the more regional one.

Ans: It is indeed a very interesting topic, but our MS mainly focuses on the spatio-temporal evolution of deformation mechanisms. The topic requires a detailed investigation including hypocenter relocation of seismicity and analysis of its spatiotemporal migration, which is beyond the scope of this study. We will prepare another paper to investigate the influence of geothermal operations on the stress accumulation and release of active faults.

Line 186. Two major issues here:

1. I would be very cautious to invert local data for global interpretation.

Ans: As you know, the coverage of InSAR data is very large with respect to the signals we are modeling. Our purpose of inverting the leveling and InSAR observations is to convince the reader that the local deformation signals are not associated with potential fault slip, rather than for global interpretation. We have performed additional tests for this revision.

2. Inversion of leveling data does not account for horizontal displacements. However, due to the local geodynamics setting strike-slip faults (e.g., the Dixieland fault?) mostly horizontal displacement would be expected.

Ans: Yes, we agree with you. For that reason the vertical and horizontal deformation observations are modelled with normal-slip feeder fault and strike-slip Dixieland fault, respectively.

Line 197. How did you measure horizontal displacements? The explanation resides in the InSAR/leveling integration. At places, this is not clear enough on the temporal evolution of the

measurements and how you inverted each period of time. How to combine both of them is sometimes vague.

Ans: Horizontal displacements are obtained through standard methods of decomposing line-of-sight InSAR maps given the presence of both ascending and descending track coverage in the same time; this gives vertical and east horizontal components. We are fortunate to have good ascending and descending coverage for this study, and the details of the processing are available in Eneva et al. (2019; GRC).

For this revision, we replotted a figure to show the temporal evolution of vertical surface displacements observed by InSAR and leveling in the main text. For the horizontal deformation, we present its temporal evolution in Figure S3.

To convince the reader that the local deformation signals are not associated with potential fault slip, we conduct three scenarios of fault slip modeling to available deformation observations. First, the long-term vertical displacement measured by leveling from 1994 to 2004 is inverted with slip on the normal-slip feeder fault. Second, the vertical transient displacement from 2006 to 2010, obtained through removing the long-term displacement trend, is also inverted with the feeder fault. Third, the horizontal displacement measured by InSAR from 2006 to 2010 is modelled with the strike-slip plate boundary (Dixieland) fault.

Line 201. Minor point: I would suggest you indicate the code used here also (although is it given in sup info)

Ans: Our numerical simulations are implemented with the software COMSOL Multiphysics (version 6.0). The model files are available from Zenodo (<https://zenodo.org/records/10460410>). We explain it in Code Availability of the revised MS based on the instructions of Nature Communications.

Line 223. “strike-slip” it is not clear which fault is which mechanism. You could introduce them better in figure 1.

Ans: The strike-slip faults include the plate boundary (Dixieland) fault and the subsidiary fault. We added two strike-slip symbols in Figure 1a.

Line 228. Why inverting for layer parameter is not feasible? Does it not depend on which ones? You need more explanations here.

Ans: The major reason is that physical processes of geothermal operations are intercoupling, dynamic and nonlinear. Further, with such a broad parameter space, achieving good results would require massive supercomputing power, which we do not have resources to use.

Figure 4. What are P1, P2, P8 etc. Please report extensively all elements within the figure caption.

Ans: Thank you for your suggestion. We add specifications to these probe points into the figure caption.

Figure 5. This figure is very unclear, mainly because the sub-figure are too small, and axis are really tiny. In addition, it is not possible to understand what are the different curves of different colors.

Ans: Our goal for this figure is to show the distinct features of pressure effects and thermal effect. During this revision, we realized that it is unnecessary to show the simulated displacements and pressure/temperature changes of all probe points in the main text, and deleted the figure. Instead, we

plot a new figure to show spatiotemporal evolution of thermo-poro-elastic effects. In the new figure, we only present the pressure effects and thermal effect at 4 probe points with good representativeness. The results of the other 4 probe points are presented in the supplementary materials.

Line 358. Figure 15 does not exist.

Ans: Sorry! It should be Figure S15.

Line 491. The initials of Jean-Philippe Avouac are J. P.

Ans: Thank you! We corrected the mistake in the revised MS.

Supplementary Materials:

Please check the use of “deformation”.

Ans: Thank you! We checked the words “deformation” and “timeseries” during this revision.

In § Fault slip inversion: I have an issue on what you want to invert for and the procedure. You remove the geothermal signal. Why and how do you do this? Then you subtract the pre-2005 trends. I really do not understand why. This may be the signal you want to invert for... You may need a reference here to make sure they are the whole area signal. After removal, what is left??

Ans: We are sorry for ambiguous introduction. During this revision, we rewrote the text on details of slip inversions. Here we would like to clarify some important points. First, the deformation signals are only observed at the HGF. Second, the surface displacement at the HGF consists of two parts: long-term component initiated before 2005 and transient component after 2005 (Fig. 3). Therefore, there are two end-member possibilities: (1) the long-term displacement associated with fault slip, and (2) the transient displacement associated with fault slip.

We thus conduct three scenarios of fault slip modeling to available deformation observations. First, the long-term vertical displacement measured by leveling from 1994 to 2004 is inverted with slip on the normal-slip feeder fault. Second, the vertical transient displacement from 2006 to 2010, obtained through removing the long-term displacement trend, is also inverted with the feeder fault. Third, the horizontal displacement measured by InSAR from 2006 to 2010 is modelled with the strike-slip plate boundary (Dixieland) fault.

in the second step and (1), you use “corrected vertical-component data”. Here 2 things: 1. what are data corrected for? The previous trends?

Ans: Yes, the data are corrected with pre-2005 (long-term) displacement trends to isolate the transient displacement signals in the vertical direction.

2. why do you invert only the vertical data? Inversion procedure with Okada model is clearly possible with all components, so I do not understand why only vertical. In addition, as the plate boundary is a strike-slip process, there may be little vertical signal but large horizontal one. So this inversion results may be completely wrong, as you invert the wrong component in which you removed some potentially valid signal... You need to review here seriously the approach.

Ans: You are correct that the Okada model provides 3D displacements; but, since our data come from different measurement techniques with different capabilities, this is a necessary limitation in some of

the tests depending on the time-period considered. Prior to 2005 we have only vertical displacements, for example. To be very clear, the vertical data is inverted with the normal-slip feeder fault. In addition, we also inverted the horizontal InSAR data with the strike-slip Dixieland fault.

In the thermo-poro-electricity formulation, in the last paragraph you indicate primary effects and secondary effects. You could list them, and justify how to classify them as primary or secondary.

Ans: Thank you for your suggestions. The three effects include: (1) influence of rock deformation on rock porosity and permeability; (2) thermal influence on fluid density and viscosity; (3) frictional heating due to rock deformation. Palciauskas and Domenico (1982) and McTigue (1986) have shown they are secondary effects.

Response Figure 8: Flowchart of two-way thermo-hydro-mechanical coupling. The arrows labeled with blue words indicate the effects included in our simulations. This can be found in the supplemental material

Figure caption of S1, S3. Check the use of “deformation” here.

Ans: Thank you. Fixed.

Figure caption of S1. The normalization method for uplift is unclear. Why do you need to normalize? Is the normalized value inverted for??

Ans: As we’ve discussed already, we only use normalized displacements for visualization and general analyses; we invert the true displacements. Further, to characterize vertical deformation trends of the HGF, we divide the study area into three regions (west, central and east) to analyze the leveling observations. The displacement time series of the 103 stations are first normalized by their root-mean-square values with the following expression,

$$n(i) = u(i) / \sqrt{\sum_{i=1}^N \frac{u(i)^2}{N-1}},$$

where $u(i)$ and $n(i)$ are the measured and normalized deformation, respectively, at each time node; and N is the total number of surveys. We then average the normalized timeseries of the stations within each sub-region to represent its deformation trend. For the whole study area with both surface uplift and subsidence, absolute values of all normalized time series are averaged to represent the deformation trend.

Figure S2. In the caption, please add the reference for the kriging here too. In the figure, you should get the same axis limits for both the data and the interpolated info.

Ans: Thank you for your suggestions. We replotted Figure S2 during this revision and added a reference for the Kriging method.

Figure S5. The sub figure are too small, making this figure useless. You also need to tell readers when... time is not indicated or clear enough.

Ans: Thank you for these comments. We replotted production rates and temperature variation of all wells of the HGF with 3 figures.

Figure S6. There are several place where the term “excess” is used for vertical displacement. I am not convince with excess it is. I did not get where this excess is coming from. Need more explanation here and before possibly.

Ans: The “excess” vertical-component displacement of 2006–2010 is obtained from removing the pre-2005 (long-term) trend. We explained it in the main text and methods for this revision.

Figure S7. Where is A-33? Show it on the figures.

Ans: The station A-33 is shown in Figure S1.

Figure S16, S17, S18. The number of sub figure is large. You could gain place by removing the axis for the inner figure, as they are the same for all. This would make the figure much easier to read. In addition, on the all 3 pages for each figure, I count 96 (4*8*3) points. Not clear where the 3 otehrs are...

Ans: Thank you for your suggestions and careful review. There are indeed 3 points that we did not present their simulation results. For this revision, we replotted Figures S16-S18.

REVIEWER COMMENTS

Reviewer #1 (Remarks to the Author):

The authors have done an excellent work revising the manuscript and all comments are addressed satisfactorily.

I do not have any further comments and recommend publishing the manuscript.

Reviewer #2 (Remarks to the Author):

This study presents an interesting case of the thermo-poro-mechanical response of the subsurface to fluid injection and production in a geothermal field. I have several concerns that are described in an attached file because I include an equation for clarity of my explanations.

[Editorial Note: This PDF is displayed across the next three pages]

Reviewer #2 (Remarks on code availability):

The readme file just states that they provide a model that can be used with COMSOL. COMSOL is a commercial software whose license is very expensive, so few people will be able to open it.

Reviewer #2 (Remarks to the Author):

Review of the manuscript “Relatively stable pressure effects and time-increasing thermal contraction control Heber geothermal field deformation”, by G. Jiang et al.

The authors have made significant changes to the manuscript in response to the reviewers' comments. However, the changes have only partially addressed the comments and new comments have arisen as a result of the modifications to the manuscript.

- 1) The authors justify the use of constant water properties because “the functions of fluid properties changing with pressure and temperature are not available”. These functions are well known and can be easily implemented in COMSOL. It is weird that the misfits shown in Table S16 as a function of water properties do not change for viscosity. The minimum and maximum tested viscosity values change by a factor of 2. Changing viscosity has a similar effect to changing permeability (in the same proportion). The misfit changes when changing reservoir permeability from 10 to 20 mD. Thus, changing viscosity by a factor of 2 should also have an effect on the misfit.
- 2) The upwards heat flow through the Feeder Fault is a variable density-driven flow in which the less dense hot water from depth migrates upwards through the Feeder Fault because of buoyancy. If water density is assumed constant, how can upwards heat flow be reliably modeled?
- 3) Figure 1 would benefit from including the direction of the maximum and minimum horizontal stresses to better understand the fault stability analysis. Apart from this, the arrows indicating the slip direction in the strike-slip faults indicate a left-lateral movement of the fault, which is the contrary to the regional movement – the San Andreas fault is a right-lateral fault. To induce a left-lateral movement of the Dixieland Fault, the maximum horizontal stress should have an azimuth around 90°, which would be perpendicular to the Feeder Fault, making it very stable (because it is a normal fault). Please, double check the orientation of the arrows indicating the movement of the strike-slip faults because they seem to have the wrong orientation. The magnitudes of the stresses used in the model should also be provided in the manuscript.
- 4) Figure 4c: after 2005, pore pressure effects have a good fitting with the leveling-observed displacement trend, so the model seems to be overestimating thermal effects. Why is there this discrepancy in the ground displacement?
- 5) Also, the observed trend in the east region, which presents an uplift followed by subsidence (Figure 3a), is not reproduced by the numerical model, which yields a continuous subsidence from the beginning of operations (Figure 8 and Supplementary Fig. 24).
- 6) Transient and rapid increases in seismicity rate are likely linked to fault reactivation and, thus, fault slip. As indicated in the text, “The most notable one started in 2005”, coinciding with the time of the reversals in the ground deformation trend. The thermo-hydro-mechanical elastic model shows that the increase in the injection rates in 2005 has an effect on the change in the trends of ground deformation, but fault reactivation and its consequent slip are likely playing a non-negligible role as well. Furthermore, the reactivated fault may continue slipping aseismically once it is reactivated. Note that surface displacement is a reflection of all the deformation that occurs below the ground and fault slip also affects ground deformation.
- 7) Fault slip modeling: it is argued that the necessary slip rate on the Feeder Fault to explain the observed ground deformation would be in the order of 4 to 10 cm/yr. This is

equivalent to 0.001 to 0.003 l.m/s – a plausible value for aseismic slip of the fault. It is argued that the slip rate is larger than the regional slip, but it should be borne in mind that injection/extraction operations induce ground deformation and may destabilize faults (see the seismicity around the faults, indicating that they are slipping). An aseismic slip rate of 0.008 l.m/s has been inferred at Brawley, California, as a result of geothermal operations (Wei et al., 2015). This slip rate is in the same order of magnitude of the one estimated in the Heber geothermal field. Thus, the hypothesis of fault slip as responsible, at least partially, of the observed ground deformation cannot be disregarded.

- 8) Another indication that faults reactivate is the observation of abrupt changes in temperature, ~ 30 °C drop after 2000. Fault slip usually enhances permeability because of dilatancy. Thus, fault reactivation and the consequent permeability enhancement could lead to a sudden increase in flow rate, affecting fluid temperature.
- 9) Specific heat of rock: typical values range from 500 to 600 J/(kg·°C), lower than the assumed value by roughly a factor of 2.
- 10) Figures 7a and 7b: it would be helpful to add dimensions to the axes.
- 11) Figure 7: we do the abrupt pressure drops, e.g., around 1994 in P4, occur in the model?
- 12) Lines 362-365: it is said that “heat transfer within the reservoir is mainly through the convection of fluid flow”. It should be indicated that it is “forced convection” because natural convection cannot be simulated in this model because fluid properties are considered constant.
- 13) Quantification of thermo-poro-elastic effects: I disagree with the way the individual effects are calculated. There are three models: (I) Biot’s coefficients equal to 0; (II) thermal expansion coefficients equal to 0; and (III) both Biot’s and thermal expansion coefficients equal to 0. Strain are proportional to effective stress changes, Δ' , which are a function of total stress changes, Δ , pore pressure changes, Δ , and temperature changes, Δ , as

$$\Delta' = \Delta - \Delta + \Delta,$$

where α is Biot’s coefficient, β is the thermal expansion coefficient and K is the bulk modulus. Thus, model III gives the poroelastic effect, i.e., the deformation as a result of the stress changes caused by pore pressure changes (and not the effect of pore pressure changes as indicated in lines 375-376). The effect of pore pressure changes can be obtained by subtracting to the calibrated model the deformation of model II and model III. And thermal effects can be obtained by subtracting to the calibrated model the deformation of model I and model III.

- 14) Discussion, lines 442-444: it is claimed that an improvement in the understanding of the spatial and temporal evolution of the strength of pore pressure and thermal effects is achieved in this study. However, the manuscript does not provide such better understanding. It is shown that there is a competition between pore pressure and thermal effects, with pore pressure presenting a rapid response and thermal effects becoming dominant in the long term, but this was already known. And it is shown that thermal contraction affects not only the cooled region, but also further away, which was already known. There is no quantification or clue how these processes can be quantified or estimated in other geothermal sites. In general, the observed coupled thermo-hydro-mechanical processes are already known. The novelty and interest of this study resides in the application to a geothermal site with data covering decades.

- 15) The two-way coupling approach is partially neglected. In particular, rock deformation is assumed to have negligible effects on temperature and pressure fields. While this may be true in porous media, it may not be like that in fractured media, like the Heber field, which has several faults. Deformation may significantly enhance fracture/fault permeability because it scales with the cube of the aperture, affecting both the pressure and temperature field.
- 16) Calibration: it is a bit surprising that the initial guess (with few exceptions of slight changes in model properties) of model properties gives the best fitting. Calibrating a 3D THM model is a challenging task.

Minor comments:

- 1) Line 341: you refer to "injection volume" but the units are given in kg
- 2) Line 561: the assumed values of fault properties are justified as "widely-used values", but the two provided references are papers of the first author of this study. References to papers by scientists that are not authoring this manuscript should be provided
- 3) Line 599: the mechanical boundary condition of "fixed injection and production wells" is not very clear. What is fixed? According to the figures in the Results, the surface displacement at the wells is not zero.
- 4) The units on both sides of Equation (3) are different: units are 1/s in the left-hand side, while they are m³/s in the right-hand side.
- 5) Line 700: please, provide references to support "typical values of faults". Faults can take wide range of values, depending on the brittleness of the rock and accumulated slip.

Reviewer #3 (Remarks to the Author):

Review of revised manuscript „Relatively stable pressure effects and time increasing thermal contraction control Heber geothermal deformation

By G. Jiang et al.

Submitted to Nature Communications

General comments

This version of the manuscript is much easier and smoother to read than the first one. The arguments are logically exposed, and the demonstration is much clearer.

However, the initial part of the argument is still a bit weak, especially concerning some terms (displacement, levelling, ...) actually many things around Figure 3, which prevent full trust in the results. Once this step passed, and admitting the observations, the demonstration is better conducted. Therefore, I strongly suggest to revise once more the manuscript to make it smooth.

Specific comments

Line 18. The term “siphonic effect” is a bit awkward here, and should be explained.

Line 19 or 20. “by cooling” could be added after “Thermal contraction “. It seems obvious, but this is clearer by adding the reason.

Figure 3 is still confusing. Fig 3a. Uz is not defined. Is it position or velocity. I do not understand why you normalize (line 104). Real values of displacement should be giving more information for possible inversion of the source location. In the west region, you show injection rate in blue. However, there are no wells in the west region, according to map Fig 3b. Where are the wells or where does this curve come from? I would add an indication in the figure 3a west, central and east of the three periods indicated below (e.g., using P1, P2 and P3, which would correspond to the periods indicated in fig 3b. I do not understand the grey zone, as it seems there is also a decade-long displacement transient from 1995 until 2005 (e.g., fig. 3a.). The term “transient” should therefore been defined. In Fig. 3b the caption indicates maps obtained from levelling and InSAR, but the top of each subfigure indicates only levelling. I understand the space is rather small to indicate both, but this is confusing.

Line 123. There is confusing reference to displacement, although you refer mostly to rate of displacement, which is velocity (for instance in this line 123, where 1.5 cm/year is indeed a velocity).

Line 124. There is inconsistency here with the main conclusion of the paper. You have both subsidence AND uplift with an increase of injection rate. Therefore, this observation contradicts line 15 of the abstracts. This needs further explanation and clarification.

Line 128-129. I do not see where in Fig 3b, levelling-measured subsidence is large than InSAR results. If this is indicated in supplementary info, you should indicate this here. Now on the result itself, I do not understand why the levelling and InSAR height changes are different. You should give a reason for this difference.

Line 135. I do not follow exactly what you mean. “... vertical displacement ... declining...” This is not very clear.

Line 138-139. I do not see this in figure 3. The west region indeed subsides before 2005, but I do see a reverse, so an increase only after 2005. However, before 2005, I do not see an increase of the injection rate. Then this sentence does not make sense. In addition, abnormal is a weird word for scientific expression. I would replace with "unexpected" or something like this. However, I do not see that in figure 3.

Line 139-140. The sentence "the same displacement anomaly..." is unclear. Which anomaly? And it is in another region and another time... so quite confusing!

Figure 4 and line 168. Is the "thermal contraction" the same as thermoelastic effect (in the figure)?

Line 171. It would be nice to precise if there any relationship between the observed horizontal displacement rate (2 cm/year) and the possibility you invoke here.

Line 320. Is it necessary to mention "Strong"?

Line 344 and 346. What is the unit "kg/mo"?

Response to Reviewer #2

The authors have made significant changes to the manuscript in response to the reviewers' comments. However, the changes have only partially addressed the comments and new comments have arisen as a result of the modifications to the manuscript.

Ans: We are really sorry that the revised MS did not completely resolve your concerns. During this revision, we have tried to address your comments as thoroughly as possible. Following are our point-by-point responses. We sincerely appreciate your careful review and critical comments.

1-1. The authors justify the use of constant water properties because “the functions of fluid properties changing with pressure and temperature are not available”. These functions are well known and can be easily implemented in COMSOL.

Ans: We only found the equation of water density changing with pressure and temperature (e.g., Wagner and Pruß, 2002),

$$\rho = \rho_0 + \frac{d\rho}{dP}(P - P_0) + \frac{d\rho}{dT}(T - T_0),$$

where ρ_0 is the reference density with the temperature of T_0 and the pressure of P_0 . However, this equation is unable to be implemented in COMSOL due to that the two items ($\frac{d\rho}{dP}$ and $\frac{d\rho}{dT}$) are not constants. For other properties of viscosity, thermal conductivity, and heat capacity, we did not find the corresponding functions.

During this revision, we learned about that the material library of COMSOL provides the changing water properties with pressure and temperature, which is obtained through interpolation. We plot figures (attached below) to show their changes with the pressure (350 to 460 K) and temperature (14 to 17 MPa) variations of the Heber geothermal reservoir. From the following figures, you can see that:

- (1) the water properties are characterized with obvious changes within the given temperature range;
- (2) the water properties are characterized with little changes within the given pressure range;
- (3) the density, thermal conductivity, and heat capacity change slightly with the pressure and temperature variation;
- (4) the viscosity has obvious changes with temperature but sharp temperature decrease is only localized near injection wells.

The material library from COMSOL was too expensive to purchase, but to resolve your concern, we used a trial version and re-ran our model with the changing water properties. The updated simulation results from 0 d to 9400 d show that the misfits to leveling, InSAR and temperature observations are 2.86 cm, 2.67 cm and 21.6°C, respectively. In contrast, the misfits of the calibrated model are 2.39 cm, 2.68 cm and 20.3°C, respectively. You can see that the model misfits become larger with the variable water properties. We tested the influence of different water properties on simulation results with results in Supplementary Table S16.

Reference: Wagner, Wolfgang, and Andreas Pruß. “The IAPWS Formulation 1995 for the Thermodynamic Properties of Ordinary Water Substance for General and Scientific Use.” *Journal of Physical and Chemical Reference Data* 31, no. 2 (2002): 387-535.

1-2. It is weird that the misfits shown in Table S16 as a function of water properties do not change for viscosity. The minimum and maximum tested viscosity values change by a factor of 2. Changing viscosity has a similar effect to changing permeability (in the same proportion). The misfit changes when changing reservoir permeability from 10 to 20 mD. Thus, changing viscosity by a factor of 2 should also have an effect on the misfit.

Ans: During this revision, we checked the model settings and found that the value of water viscosity in our model was fixed to be the initial value of 2.5E-4 Pa*s. Thank you very much for careful review! We corrected this error and re-ran our simulations with the test water properties. Table S16 (we attach it below) lists the updated results for the model misfits, which are still smallest with the initial values of water properties.

RMS misfit	Density (kg/m ³)			Viscosity (Pa·s)				Compressibility (Pa ⁻¹)		
	800	900	1000	1.5×10 ⁻⁴	2×10 ⁻⁴	2.5×10 ⁻⁴	3×10 ⁻⁴	3.7×10 ⁻¹⁰	4.2×10 ⁻¹⁰	4.7×10 ⁻¹⁰
Leveling (cm)	2.42	2.65	2.39	3.39	2.57	2.39	3.03	3.67	2.39	3.18
InSAR (cm)	3.39	2.60	2.68	2.58	2.84	2.68	2.59	4.20	2.68	2.79
Weighted Misfit (cm)	2.61	2.64	2.45	3.23	2.62	2.45	2.94	3.78	2.45	3.10
Temperature (°C)	20.36	20.30	20.30	20.69	20.42	20.30	21.21	20.42	20.30	20.96
RMS misfit	Thermal Conductivity (W/(m·°C))			Specific Heat (J/(kg·°C))						
	0.58	0.68	0.78	3900	4200	4500	4800			
Leveling (cm)	2.39	2.39	2.39	2.66	2.39	2.62	2.80			
InSAR (cm)	2.68	2.68	2.68	2.51	2.68	2.41	2.99			
Weighted Misfit (cm)	2.45	2.45	2.45	2.63	2.45	2.58	2.84			
Temperature (°C)	20.30	20.30	20.30	20.55	20.30	20.53	20.44			

2. The upwards heat flow through the Feeder Fault is a variable density-driven flow in which the less dense hot water from depth migrates upwards through the Feeder Fault because of buoyancy. If water density is assumed constant, how can upwards heat flow be reliably modeled?

Ans: According to Lippman and Bodvarsson' studies (attached below), exploration wells verified the presence of significant hydrothermal flow through the feeder fault zone, with temperatures of nearly 245°C and rates around 14.6 kg/s (see the following conceptual figure from Lippman and Bodvarsson, 1985). To identify the replenishment mechanism of heat flow at the HGF, we conducted additional simulations under two scenarios: (I) one upwelling channel of the feeder fault with different rates of heat flux set as a boundary condition at the base of the fault; (II) two channels of the feeder and subsidiary strike-slip faults with different flow rates.

1. Lippman, M.J., and Bodvarsson, G.S. (1983). A modeling study of the natural state of the Heber Geothermal Field, California. *Geothermal Resources Council Trans.*, Vol. 7, p. 441-447.
2. Lippman, M.J., and Bodvarsson, G.S. (1983). The generating capacity of the Heber Geothermal Field, California. *Proceedings Ninth Workshop Geothermal Reservoir Engineering*, p. 157-166.
3. Lippman, M.J., and Bodvarsson, G.S. (1985). The Heber geothermal field, California: natural state and exploitation modeling studies. *Journal of Geophysical Research*, 90 (B1), 745-758, doi:10.1029/JB090iB01p00745.

[REDACTED]

3-1. Figure 1 would benefit from including the direction of the maximum and minimum horizontal stresses to better understand the fault stability analysis.

Ans: Thank you very much for your suggestion! We add the arrows into Figure 1a (we attached it below) to show the orientation of regional background principal stress.

[REDACTED]

The stress orientation is from the SCEC Community Stress Model (<https://www.scec.org/research/csm>), which compiles multiple sources of information.

3-2. Apart from this, the arrows indicating the slip direction in the strike-slip faults indicate a left-lateral movement of the fault, which is the contrary to the regional movement – the San Andreas fault is a right-lateral fault. To induce a left-lateral movement of the Dixieland Fault, the maximum horizontal stress

should have an azimuth around 90° , which would be perpendicular to the Feeder Fault, making it very stable (because it is a normal fault). Please, double check the orientation of the arrows indicating the movement of the strike-slip faults because they seem to have the wrong orientation.

Ans: Thank you very much for your careful review! Yes, we plotted the arrows with wrong orientations. The two strike-slip faults in our study region are indeed right-lateral. We corrected the error in Figure 1a.

3-3. The magnitudes of the stresses used in the model should also be provided in the manuscript.

Ans: In our model, we consider the field to be in stress equilibrium at time zero, and did not consider the regional background stress. Alternatively, the mechanical boundary conditions include: (1) free upper surface, (2) roller boundaries for the bottom and side surfaces, and (3) fixed open-hole sections of injection and production wells. In addition, we used the simulated stress at $t=0$ to make the model achieve initial stress balance, and therefore the model simulates the impacts of geothermal operations on the state of stress and fluid pressure and temperature.

4. Figure 4c: after 2005, pore pressure effects have a good fitting with the leveling observed displacement trend, so the model seems to be overestimating thermal effects. Why is there this discrepancy in the ground displacement?

Ans: Leveling-observed surface displacement trend and temporal growth of seismicity within the HGF is highly correlated with the normalized vertical displacements linked to thermal contraction (Fig. 4c), with the correlation coefficients up to 0.85 and 0.97, respectively. In contrast, the coefficients of the leveling displacements and seismicity growth relative to the normalized displacements associated with the pressure effects are only 0.09 and 0.23, respectively.

During this revision, we further calculate the correlation coefficients for the time period after 2005. The coefficient between the leveling observations after 2005 and the contribution of pore pressure effects is -0.02. In contrast, the coefficient between the leveling observations after 2005 and the contribution of thermal effect is 0.89.

5. Also, the observed trend in the east region, which presents an uplift followed by subsidence (Figure 3a), is not reproduced by the numerical model, which yields a continuous subsidence from the beginning of operations (Figure 8 and Supplementary Fig. 24).

Ans: Yes, our model cannot simulate the ground uplift of the east region before 2005; but our model does not overfit the subsidence after 2005 (see the following two figures: one shows the locations of leveling benchmarks; another shows the observed and simulated leveling time series). Therefore, we do not think that our model overestimates the thermal effect. It is hard for us to present potential reasons for the bad fit before 2005 aside from the obvious sources of uncertainty related to reservoir structure. However, based on our modeling we gained the following insights: (1) the observed surface displacements far outside the HGF scope are harder to be fit than the observations within the scope with geothermal operations; and (2) the model parameters calibrated with available observations represent the rock properties in average. For the peripheral region of the HGF without geothermal operations, the displacement observations are prone to be disturbed by other physical processes like natural recharge of regional groundwater. We note this directly in the text.

6. Transient and rapid increases in seismicity rate are likely linked to fault reactivation and, thus, fault slip. As indicated in the text, “The most notable one started in 2005”, coinciding with the time of the reversals in the ground deformation trend. The thermohydro-mechanical elastic model shows that the increase in the injection rates in 2005 has an effect on the change in the trends of ground deformation, but fault reactivation and its consequent slip are likely playing a non-negligible role as well. Furthermore, the reactivated fault may continue slipping aseismically once it is reactivated. Note that surface displacement is a reflection of all the deformation that occurs below the ground and fault slip also affects ground deformation.

Ans: Thank you for your comments! But there is no robust evidence to support the hypothesis that aseismic fault slip plays an important role in surface deformation, and we carefully tested this possibility in our analyses. In addition to your note, we also note that the transient displacement seen in the vertical direction is dominated by uplift, and our tests show that such uplift cannot be fit by reasonable slip on the normal-slip feeder fault (see the supplementary Fig. 7).

Supplementary Fig. 7. Fault modeling of vertical transient displacement from 2006 to 2010. a Corrected leveling displacement with removing the long-term displacement trends observed before 2005. **b** Misfit between leveling observations and predictions with the inverted slip distribution on the normal-slip feeder fault shown in **c** (2D view) and **d** (3D view). Negative value represents normal slip.

7. Fault slip modeling: it is argued that the necessary slip rate on the Feeder Fault to explain the observed ground deformation would be in the order of 4 to 10 cm/yr. This is equivalent to 0.001 to 0.003 $\mu\text{m/s}$ – a plausible value for aseismic slip of the fault. It is argued that the slip rate is larger than the regional slip, but it should be borne in mind that injection/extraction operations induce ground deformation and may destabilize faults (see the seismicity around the faults, indicating that they are slipping). An aseismic slip rate of 0.008 $\mu\text{m/s}$ has been inferred at Brawley, California, as a result of geothermal operations (Wei et al., 2015). This slip rate is in the same order of magnitude of the one estimated in the Heber geothermal field. Thus, the hypothesis of fault slip as responsible, at least partially, of the observed ground deformation cannot be disregarded.

Ans: This is a good comment, but we do not find supporting evidence for this. Because the surface displacement at the HGF consists of two parts – long-term displacement initiated before 2005 and transient displacement after 2005 – we conducted three scenarios of fault slip modeling of the available geodetic observations. First, the long-term vertical displacement measured by leveling from 1994 to 2004 is inverted with slip on the normal-slip feeder fault. Second, the vertical transient displacement from 2006 to 2010, obtained through removing the long-term displacement trend, is also inverted with

the feeder fault. Third, the horizontal displacement measured by InSAR from 2006 to 2010 is modelled with the strike-slip plate boundary (Dixieland) fault.

Our inversions show that: (1) vertical transient displacement cannot be fit by slip on the feeder fault; (2) although the long-term vertical displacement can be fit by slip on the feeder fault, the slip rates are required to be 4~10 cm/yr. If such slip exist, the cumulative slip on the feeder fault is up to 1.2~3 m during 30 years of geothermal operation. Further, as we presented in Figure 2 (attached below), there are several wells intersecting with the feeder fault which are the main production wells for the geothermal plant. If long-term aseismic slip exists, such large cumulative slip would cause catastrophic, irreversible casing deformation of the wells intersecting with the feeder fault; but there were no related reports on such damage and production rates have remained relatively constant, implying that not major repairs to the production wells in response to fault slip has been needed. We note this in the text.

8. Another indication that faults reactivate is the observation of abrupt changes in temperature, $\sim 30\text{ }^{\circ}\text{C}$ drop after 2000. Fault slip usually enhances permeability because of dilatancy. Thus, fault reactivation and the consequent permeability enhancement could lead to a sudden increase in flow rate, affecting fluid temperature.

Ans: We present the temperature changes of all wells in Figure 2 above. We believe you are referring to the abrupt changes of the wells in the east region around 2006. All the abrupt changes can be predicted by our model based on injection and production alone (see Figure 3 below). In other words, we show that the temperature variations are associated with geothermal operations with no need to invoke fault permeability changes.

9. Specific heat of rock: typical values range from 500 to 600 J/(kg·°C), lower than the assumed value by roughly a factor of 2.

Ans: According to Table 12 from Robertson E. C. (1988), the value of specific heat varies from 800 J/(kg·°C) to 1200 J/(kg·°C) with the temperature increasing from 50°C to 200°C. We note this in the text.

[REDACTED]

Robertson E. C. (1988), Thermal properties of rocks, USGS Open-File Report 88-441.

10. Figures 7a and 7b: it would be helpful to add dimensions to the axes.

Ans: Thank you for your suggestion! We revised Figures 7a and 7b.

11. Figure 7: we do the abrupt pressure drops, e.g., around 1994 in P4, occur in the model?

Ans: The abrupt pressure changes are totally caused by geothermal operations. Again, we are using the injection and production values reported by the geothermal operator as inputs to the model.

12. Lines 362-365: it is said that “heat transfer within the reservoir is mainly through the convection of fluid flow”. It should be indicated that it is “forced convection” because natural convection cannot be simulated in this model because fluid properties are considered constant.

Ans: OK, we revised the sentence. Thank you for your suggestion!

13. Quantification of thermo-poro-elastic effects: I disagree with the way the individual effects are calculated. There are three models: (I) Biot's coefficients equal to 0; (II) thermal expansion coefficients equal to 0; and (III) both Biot's and thermal expansion coefficients equal to 0. Strain are proportional to effective stress changes, $\Delta\sigma'$, which are a function of total stress changes, $\Delta\sigma$, pore pressure changes, Δp , and temperature changes, ΔT , as $\Delta\sigma' = \Delta\sigma - \alpha\Delta p + \beta K\Delta T$, where α is Biot's coefficient, β is the thermal expansion coefficient and K is the bulk modulus. Thus, model III gives the poroelastic effect, i.e., the deformation as a result of the stress changes caused by pore pressure changes (and not the effect of pore pressure changes as indicated in lines 375-376). The effect of pore pressure changes can be obtained by subtracting to the calibrated model the deformation of model II and model III. And thermal effects can be obtained by subtracting to the calibrated model the deformation of model I and model III.

Ans: Thank you very much for your comments! During this revision, we checked a classic reference on the topic, “*Theory of Linear Poroelasticity with Applications to Geomechanics and Hydrogeology*” written by H. F. Wang, where we found the equation of total strain of rock skeleton (see the attached screenshot for the expression).

The three additional equations, which relate the shear strains to the shear stresses, are

$$\epsilon_{xy} = \frac{1}{2G} \sigma_{xy} \quad (2.33)$$

$$\epsilon_{yz} = \frac{1}{2G} \sigma_{yz} \quad (2.34)$$

$$\epsilon_{xz} = \frac{1}{2G} \sigma_{xz} \quad (2.35)$$

Eqns. 2.33–2.35 do not contain a pore pressure term because changes in pore pressure are assumed not to induce shear strain. Eqns. 2.30–2.35 can all be represented by a single equation in index notation:

$$\underbrace{\epsilon_{ij}}_{\text{Total Strain}} = \underbrace{\frac{1}{2G} \left[\sigma_{ij} - \frac{\nu}{1+\nu} \sigma_{kk} \delta_{ij} \right]}_{\text{Poroelastic Strain}} + \underbrace{\frac{\alpha}{3K} p \delta_{ij}}_{\text{Free Strain}} \quad (2.36)$$

where the Kronecker delta in Eqn. 2.36 is defined by

$$\delta_{ij} = \begin{cases} 1 & \text{if } i = j \\ 0 & \text{if } i \neq j \end{cases} \quad (2.37)$$

From equation 2.36 from Wang's book, we can see that the total strain consists of two parts: poroelastic strain and free strain. The free strain is associated with pressure changes. Therefore, the equation of total strain of our reference model can be simplified as:

$$\mathcal{E}_{Total} = \mathcal{E}_{Poroelastic} + \mathcal{E}_{Pressure} + \mathcal{E}_{Thermal}$$

The corresponding equations of models I, II, and III can be simplified as:

$$\text{I: } \mathcal{E}_{Total} = \mathcal{E}_{Poroelastic} + \mathcal{E}_{Thermal}$$

$$\text{II: } \mathcal{E}_{Total} = \mathcal{E}_{Poroelastic} + \mathcal{E}_{Pressure}$$

$$\text{III: } \mathcal{E}_{Total} = \mathcal{E}_{Poroelastic}$$

We agree with you that “model III gives the poroelastic effect”; but the effect of pore pressure changes can be obtained by comparison of the reference model with model I. The thermal effects can be isolated through comparison of the reference model with model II. In our MS, the poroelastic contribution was mistaken as the pressure contribution; and the pressure contribution was mistaken as the poroelastic contribution. We corrected the error during this revision.

14. Discussion, lines 442-444: it is claimed that an improvement in the understanding of the spatial and temporal evolution of the strength of pore pressure and thermal effects is achieved in this study. However, the manuscript does not provide such better understanding. It is shown that there is a competition between pore pressure and thermal effects, with pore pressure presenting a rapid response and thermal effects becoming dominant in the long term, but this was already known. And it is shown

that thermal contraction affects not only the cooled region, but also further away, which was already known. There is no quantification or clue how these processes can be quantified or estimated in other geothermal sites. In general, the observed coupled thermo-hydromechanical processes are already known. The novelty and interest of this study resides in the application to a geothermal site with data covering decades.

Ans: Respectfully, we disagree with these characterizations of the paper. We are not aware of previous studies that indicate that all of this ‘was already known’ in a generalized way. There are theoretical models for all of these phenomena, of course, but we have not claimed to invent new theory. In practice there are geodetic observations from almost every active geothermal field, or course, but this is the first study to have decades of time-varying surface displacement changes with advanced modeling capable of replicating the complex signals seen in time in space. Fundamentally, our prior understanding of deformation and seismicity at geothermal fields may be limited if not biased by the fact that most sites do not have such wealth of information. In other words, while the individual phenomena considered here have each been proposed to explain limited geodetic observations in the past, it is clear from this study that multiple phenomena act in concert. This finding would not have been possible in practice without long-term geodetic observations, and thus we will need far more cases where decades of geodetic data are available to establish general patterns. With that said, the modeling approach taken can be adopted at any geothermal field, which we note in the text.

Below we provide a little more discussion on what makes this study unique:

As you know that deformation of geothermal fields can be caused by reservoir pressure depletion, poroelastic and thermoelastic contraction. Such mechanisms can be active concurrently, but it is challenging to quantify their relative strengths and spatiotemporal evolution features, especially for large-scale fields with complex operation history. One major reason is due to the rarity of long-term seismic and geodetic observations combined with the extreme difficulty of setting up a realistic 3D thermo-hydro-geological model with limited subsurface geological and geophysical data (like reservoir geometry, formation lithology and fault distribution) needed to simulate decades of geothermal operations.

In this study, we quantify spatiotemporal evolution of the relative strengths of the three physical mechanisms for the Heber Geothermal Field (HGF) based on a 3D thermo-hydro-geological model, which is calibrated by multiple datasets, including geodetic observations over 20 years, precise well trajectories and operational records of fluid rates and temperatures. Our most important findings are:

- 1. The effects of pressure fluctuation and poroelastic response keep relatively stable in the operation history of the HGF. In contrast, the strength of thermal contraction effect increases with time: after ~30 years of operations, the thermal effect is 2-3 times larger than the two pressure effects.**
- 2. It is common that fluid injection causes surface uplift due to pore pressure increases within the target reservoir; however, the ground in west and east regions of the HGF was subsiding during some time periods with net increases of injected fluid. We show that the deformation anomaly is related to the effect of the HGF center siphoning fluid from surrounding regions and ever-increasing effect of thermal contraction.**

3. The long-term trends of surface deformation and seismicity growth at the HGF are controlled by the thermal effect. In contrast, the two pressure effects only drive instantaneous changes.

The finding that pressure and thermal effects dominate the early- and late-stage deformation of geothermal fields, respectively, helps to reconcile the controversy of differing deformation mechanisms reported for the same geothermal field, like at the Coso field (*Fialko and Simons, 2000; Im et al., 2021*) and the Brady Hot Springs field (*Reinisch et al., 2020; Ali et al., 2016*). In addition, the framework, developed for resolving the spatiotemporal evolution of the HGF deformation mechanisms through integrating multiple geodetic, geophysical and geological data, is applicable to other complex geothermal systems.

15. The two-way coupling approach is partially neglected. In particular, rock deformation is assumed to have negligible effects on temperature and pressure fields. While this may be true in porous media, it may not be like that in fractured media, like the Heber field, which has several faults. Deformation may significantly enhance fracture/fault permeability because it scales with the cube of the aperture, affecting both the pressure and temperature field.

Ans: Thank you for this comment. We agree, in principle, and it is true that the cubic flow law is a well known empirical relationship between fracture aperture and hydraulic conductivity. However, incorporating this level of detail in our model would dramatically increase both the computational burden and the parameter space, and it is unclear that the added complexity would yield a ‘better’ result. It is also extremely challenging and uncertain to try and characterize the fracture system without significant relevant data, which we do not have. If we gained access to fluid pressure and temperature observations from within the reservoir, we could test for various flow regimes and their effect on the deformation field; unfortunately these are proprietary and likely never to be shared publicly.

During our model calibration, we test the influences of different properties (mechanical, hydrogeological and thermophysical parameters) of the geothermal reservoir and faults on simulation results. These simulations reflect to some extent the influence of rock deformation on rock properties. In addition, our model calibration shows that the influences of hydraulic properties and fault thickness are largest, indicating that surface displacement and heat transfer of the HGF are mainly associated with fluid flow, and that the influence of rock properties is relatively minor. In our study, we focus on the dominant deformation mechanisms of the Heber geothermal field. Therefore, the coupling between rock deformation on temperature and pressure is not considered in our study.

16. Calibration: it is a bit surprising that the initial guess (with few exceptions of slight changes in model properties) of model properties gives the best fitting. Calibrating a 3D THM model is a challenging task.

Ans: Yes, we totally agree! This work is based on more than two years of calibration work including: (1) designing and adjusting the calibration strategies; (2) adjusting the initial parameter settings based on prior studies on the Heber geothermal field, and the properties of water and rock. One can think of the starting parameters as the most informed guess using all the available constraints we could find, and the calibration procedure as testing variations from that set of parameters.

Minor comments:

1) Line 341: you refer to “injection volume” but the units are given in kg

Ans: Thank you very much for your careful review! We revised the sentence to be “*In the western region of the HGF, although the net injection rates are around 0.6×10^9 kg per month on average from 1994 to 2005*”.

2) Line 561: the assumed values of fault properties are justified as “widely-used values”, but the two provided references are papers of the first author of this study. References to papers by scientists that are not authoring this manuscript should be provided

Ans: Thank you very much for your suggestion. We add three papers of Chang and Segall (2016), Zbinden et al. (2020), and De Simone et al. (2013) into our MS as relevant references:

1. Chang, K. W., and P. Segall (2016), Injection-induced seismicity on basement faults including poroelastic stressing, *J. Geophys. Res. Solid Earth*, 121, 2708–2726, doi:10.1002/2015JB012561.
2. Zbinden, D., Rinaldi, A. P., Diehl, T., & Wiemer, S. (2020). Hydromechanical modeling of fault reactivation in the St. Gallen deep geothermal project (Switzerland): Poroelasticity or hydraulic connection? *Geophysical Research Letters*, 47, e2019GL085201. <https://doi.org/10.1029/2019GL085201>
3. De Simone, S., V. Vilarrasa, J. Carrera, A. Alcolea, and P. Meier (2013), Thermal coupling may control mechanical stability of geothermal reservoirs during cold water injection, *Phys. Chem. Earth*, 64, 117–126.

3) Line 599: the mechanical boundary condition of “fixed injection and production wells” is not very clear. What is fixed? According to the figures in the Results, the surface displacement at the wells is not zero.

Ans: Thank you very much for your careful review! The open-hole sections of all wells are fixed to the material layers in our model. This implies there would be some deformation of the open hole section as the surrounding material deforms. We revised the sentence in MS.

4) The units on both sides of Equation (3) are different: units are 1/s in the left-hand side, while they are m³/s in the right-hand side.

Ans: Thank you for your careful review! Again we checked the equations in H.F. Wang’s book (relevant section attached below). In equations 4.65 and 4.66 from that book, the term Q presents the volume of fluid per unit bulk volume per unit time. In our equation, Q_m presents the corresponding mass source. Our equation is consistent with the equation 4.66 from Wang (2000).

4.7.2 Pore Pressure—Volumetric Strain Equation

As was the case for the constitutive equations, either pore pressure or increment of fluid content can be chosen to be the fluid variable, and stress or strain can be chosen to be the mechanical variable (see Table 2.1) in the fluid diffusion equation. Eqn. 4.64 used pore pressure and mean stress as the fluid and mechanical variables, respectively. The permutation considered in this section is pore pressure and volumetric strain.

Eqn. 2.24 is substituted into Eqn. 4.64 to yield

$$\alpha \frac{\partial \epsilon_{kk}}{\partial t} + S_{\epsilon} \frac{\partial p}{\partial t} = \frac{k}{\mu} \nabla^2 p + Q \quad (4.65)$$

where S_{ϵ} is the specific storage at constant strain (cf. Eqn. 3.40). Eqn. 4.65 is, again, an inhomogeneous diffusion equation for pore pressure. The pore pressure field is coupled with the time rate of change of the volumetric strain. This term behaves mathematically like a fluid source.

4.7.3 Hydrogeologic Transient Flow Equation

The assumptions of uniaxial strain and constant vertical stress led to the simplified constitutive relation $\zeta = Sp$ (Eqn. 3.49), which when substituted into Eqn. 4.62 yields the standard transient flow equation used in hydrogeology:

$$S \frac{\partial p}{\partial t} = \frac{k}{\mu} \nabla^2 p + Q \quad (4.66)$$

5) Line 700: please, provide references to support “typical values of faults”. Faults can take wide range of values, depending on the brittleness of the rock and accumulated slip.

Ans: This is a great point. We took the approach of assuming that because the faults are either high-conductivity flow zones (feeder fault) or plate boundary faults, that they should have reduced elastic moduli in the damage zone. You are correct there is significant variations possible. We based the initial values for the surrounding rock on the seismic velocity model converted to elastic parameters, and then reduced the moduli of the fault zones accordingly.

Response to Reviewer #3

This version of the manuscript is much easier and smoother to read than the first one. The arguments are logically exposed, and the demonstration is much clearer. However, the initial part of the argument is still a bit weak, especially concerning some terms (displacement, levelling, ...) actually many things around Figure 3, which prevent full trust in the results. Once this step passed, and admitting the observations, the demonstration is better conducted. Therefore, I strongly suggest to revise once more the manuscript to make it smooth.

Ans: Thank you again for reading through our manuscript carefully. We have made further revisions based on the comments from you and Reviewer #2. Following are our point-by-point responses to your specific comments.

1. Line 18. The term “siphonic effect” is a bit awkward here, and should be explained.

Ans: Thank you for your suggestion. We should have referred to this as ‘siphoning’ rather than ‘siphonic’. To be clear, the siphoning effect refers the phenomenon introduced in Line 14: *high-yield production wells at the HGF center siphon fluid from surrounding regions*. During this revision we are unable to elaborate in the abstract because of tight word limits (150 words). But, in the text we use the definite article “the” to indicate the phenomenon described in Line 14, and fix the phrasing.

2. Line 19 or 20. “by cooling” could be added after “Thermal contraction “. It seems obvious, but this is clearer by adding the reason.

Ans: Thank you for your suggestion. We added the words “by cooling” into Line 17, where the words “thermal contraction” first appear.

3-1. Figure 3 is still confusing. Fig 3a. Uz is not defined. Is it position or velocity.

Ans: Thank you for your careful review! “Uz” is vertical displacement. We add the explanation into the caption.

3-2. I do not understand why you normalize (line 104). Real values of displacement should be giving more information for possible inversion of the source location.

Ans: The displacement time series of the leveling stations are normalized for two purposes primarily related to visualization: (1) to represent the deformation trends of the west, central and east zones of the HGF, which have similar temporal characteristics but at different scales, and (2) to represent temporal variation of deformation of the whole region. By doing this, we can compare vertical deformation trends of the west, central and east regions with the respective net injection volumes of geothermal fluid of every month (Figure 3a). In addition, we can also compare the deformation trend of the whole region with temporal changes of observed seismicity in the Heber geothermal field (Figure 4).

To be clear, the true values of displacement were used for model calibration, and misfits in both the hydro-geo-mechanical simulations and the fault slip inversions were calculated with actual observations and model predictions. We simplified the text to eliminate any confusion about how the normalized observations are used.

3-3. In the west region, you show injection rate in blue. However, there are no wells in the west region, according to map Fig 3b. Where are the wells or where does this curve come from?

Ans: Thank you for your comment! As we introduced in Line 65, there are 28 injection and 31 extraction wells as well as 10 wells that were first operated for ~2 years of heat extraction, and then used for reinjecting heat-depleted brines since 1993. From Figure 1a and Figure 2 (we attach it below), you can see that there are only extraction wells in the west and east regions. The panels in Figure 3b are characterized with limited space; if we plot all wells in one panel, there will be too much clutter and overlap (as shown in Figure 1a). Consequently, we plot the injection wells, extraction wells, and extraction-injection wells in three panels of Figure 3b, respectively. See response to your comment #3-6 for Figure 3b.

3-4. I would add an indication in the figure 3a west, central and east of the three periods indicated below (e.g., using P1, P2 and P3, which would correspond to the periods indicated in fig 3b).

Ans: We are wondering if there is a misunderstanding of Figure 3a. The three panels in Figure 3a show the vertical displacement trends of the west, central, and east zones of the Heber geothermal field.

3-5. I do not understand the grey zone, as it seems there is also a decade-long displacement transient from 1995 until 2005 (e.g., fig. 3a.). The term “transient” should therefore be defined.

Ans: Thank you for your suggestion! We defined the “transients” in the MS as: *The displacement rate and trend changes that initiated around 2005 and lasted for more than 10 years are referred to as “decade-long transients”*.

3-6. In Fig. 3b the caption indicates maps obtained from levelling and InSAR, but the top of each subfigure indicates only levelling. I understand the space is rather small to indicate both, but this is confusing.

Ans: In Figure 3b (we attach it below), there are 3 panels. The first panel shows the vertical displacement rate map derived from interpolation of leveling observations from January 1994 to December 2004. The last two panels show the vertical displacement rate map derived from interpolation of leveling and InSAR observations from 2006 to 2010. The purpose to show the InSAR map is to make a comparison with the leveling results during a comparable time-period.

4. Line 123. There is confusing reference to displacement, although you refer mostly to rate of displacement, which is velocity (for instance in this line 123, where 1.5 cm/year is indeed a velocity).

Ans: Thank you for your comments, but we think that “displacement rate” are also common words and are widely used without any potential ambiguities.

5. Line 124. There is inconsistency here with the main conclusion of the paper. You have both subsidence AND uplift with an increase of injection rate. Therefore, this observation contradicts line 15 of the abstracts. This needs further explanation and clarification.

Ans: Thank you for your comment! We revised the sentence of Line 15 to be “*Here we show that high-yield production wells at the HGF center siphons fluid from surrounding regions, which can cause subsidence at low-rate injection locations.*”.

6. Line 128-129. I do not see where in Fig 3b, levelling-measured subsidence is large than InSAR results. If this is indicated in supplementary info, you should indicate this here. Now on the result itself, I do not understand why the levelling and InSAR height changes are different. You should give a reason for this difference.

Ans: From the last two panels of Figure 3b (we attach it above), you can see that the leveling-measured subsidence at the HGF center is larger than the InSAR result. These are, of course, different geodetic methods subject to different influences and sources of error and different resolutions in both time and space. Hence, one possible reason relates to velocities for both datasets being derived from linear regression for the period indicated. As the leveling shows, the rate of change in the vertical direction is non-linear, which suggests there would be some bias between the two vertical results, since there are different covariance structures and different number of data. We added a brief explanation into the main text.

References: Eneva, M., Adams, D., Hsiao, V., Falorni, G. and Locatelli, R., 2019. Surface Deformation at the Heber Geothermal Field in Southern California. In Proceedings of the 44th Workshop on Geothermal Reservoir Engineering, SGP-TR-214. Stanford University, Stanford, California, February (pp. 11-13).

7. Line 135. I do not follow exactly what you mean. “... vertical displacement ... declining...” This is not very clear.

Ans: In the first panel of Figure 3a (we attach it below), you can see that the vertical displacement is characterized by subsidence before 2005 and uplift after 2005. The temporal variation trend (red error bars) is similar to the changes of injection rates (blue curve). During this revision, we rewrote the

sentence to be “the vertical displacement is characterized by subsidence before 2005 and uplift after 2005 with the temporal variations which are similar to the changes of injection rates”.

8. Line 138-139. I do not see this in figure 3. The west region indeed subsides before 2005, but I do see a reverse, so an increase only after 2005. However, before 2005, I do not see an increase of the injection rate. Then this sentence does not make sense. In addition, abnormal is a weird word for scientific expression. I would replace with “unexpected” or something like this. However, I do not see that in figure 3.

Ans: From 1994 to 2005, both the vertical displacement and injection rates are characterized by a declining trend. As you know, fluid injection is expected to cause surface uplift. Consequently, it is an unexpected phenomenon that the ground of the west region was subsiding from 1994 to 2005 even with a net increase of injected fluid. During this revision, we replaced the word “abnormal” with “unexpected”.

9. Line 139-140. The sentence “the same displacement anomaly...” is unclear. Which anomaly? And it is in another region and another time... so quite confusing!

Ans: In the third panel of Figure 3a (we attach it below), you can see that the ground of the east region was subsiding from 2005 to 2017 with a net increase of injected fluid. This is the displacement anomaly that we want to highlight. During this revision, we replaced the sentence to be “In the east region, the ground was also subsiding from 2005 to 2017 with a net increase of injected fluid”.

10. Figure 4 and line 168. Is the “thermal contraction” the same as thermoelastic effect (in the figure)?

Ans: Yes, they are the same. During this revision, we replaced the words “thermoelastic effect” in Figure 4 with “thermal contraction”.

11. Line 171. It would be nice to precise if there any relationship between the observed horizontal displacement rate (2 cm/year) and the possibility you invoke here.

Ans: According to our modeling, for a distributed slip model to even come close to matching the InSAR-measured horizontal displacement from 2006 to 2010, the magnitudes of dextral strike-slip on the plate boundary fault need to be up to 30 to 82 cm below the depth of 8 km (Supplementary Fig. 8), which suggests 6 to 16 cm/yr of slip rates. If that were true, the slip rates would be much higher than the reported slip rates and the relative plate motion rate (only ~4 cm/yr), which cannot be reconciled. Consequently, the dominant reason for the horizontal displacement is unlikely related to fault slip.

12. Line 320. Is it necessary to mention “Strong”?

Ans: We know that heat convection associated with fluid flow occurs everywhere within the geothermal reservoir. But the convection is stronger in some localized regions with higher porosity and permeability, which can cause prominent temperature decrease. So we think it is necessary to keep the word “strong”.

13. Line 344 and 346. What is the unit “kg/mo”?

Ans: “kg” is the abbreviation for “kilogram”; “mo” is the most common abbreviation for “month”.

REVIEWERS' COMMENTS

Reviewer #2 (Remarks to the Author):

Review of the manuscript “Thermal contraction gradually dominates Heber geothermal field deformation”, by Jiang et al.

The authors have corrected some errors that they had in figures and in the description of the methods (see, for example, response to comment 13 of reviewer 2). The results have not changed, so I hope that the error was just in the description and that the analysis was performed correctly. There are some aspects in which the authors and I continue to disagree, like the origin of the needed slip rate of the Feeder Fault to explain the observed deformation (I have explained, giving reference values from other sites, that a slip rate of 4-10 cm/yr can be achieved with aseismic slip of the fault). I do not think that such disagreement in the interpretation of the causes of the ground deformation should prevent this manuscript from being published, I think it is normal that we have different views, especially in complex sites like the studied one. Apart from this, I think there is already a quite good understanding of thermal effects and their temporal evolution with respect to poromechanical effects. I recall a figure from the report on induced seismicity of the National Academy of Sciences (2013) showing production and injection rates at The Geysers, together with monitored induced seismicity. Low-magnitude seismicity correlates with injection volumes, i.e., the poromechanical response is immediate and mainly local, affecting fractures in the vicinity of wells. Apart from this, there is an increase in the frequency of magnitude 3 earthquakes after 20 years from the start of water injection, and of magnitude 4 earthquakes after 30 years. This shows that the thermomechanical response takes time and that it extends further away than the cooled region, destabilizing distant faults. Similar observations have been shown through numerical simulations of geothermal systems. This is why I was saying that the explained processes at HGF are already known.

Response to Reviewer #2

The authors have corrected some errors that they had in figures and in the description of the methods (see, for example, response to comment 13 of reviewer 2). The results have not changed, so I hope that the error was just in the description and that the analysis was performed correctly.

Ans: Thank you very much for your comment. During last revision, we carefully checked our model and simulation results based on the comments from you and Reviewer #3. We found that the values of water properties in our model were fixed to their initial values and that the strengths of poroelastic effect was mistaken as the pressure effect. We corrected these errors during last revision; they did not change the conclusions of the paper.

There are some aspects in which the authors and I continue to disagree, like the origin of the needed slip rate of the Feeder Fault to explain the observed deformation (I have explained, giving reference values from other sites, that a slip rate of 4-10 cm/yr can be achieved with aseismic slip of the fault). I do not think that such disagreement in the interpretation of the causes of the ground deformation should prevent this manuscript from being published, I think it is normal that we have different views, especially in complex sites like the studied one.

Ans: We totally understand your concern, and appreciate your perspective. However, if the slip rates on the Feeder fault are indeed 4~10 cm/yr, the cumulative slip would be 1.2~3 m during 30 years of geothermal operation. If this kind of long-term aseismic slip exists, such large cumulative slip would cause catastrophic, irreversible casing/borehole deformation of the wells intersecting with the feeder fault; but there were no related reports on such damage and production rates have remained relatively constant, implying that major repairs to the production wells (or re-drilling) in response to fault slip has not been needed.

Apart from this, I think there is already a quite good understanding of thermal effects and their temporal evolution with respect to poromechanical effects. I recall a figure from the report on induced seismicity of the National Academy of Sciences (2013) showing production and injection rates at The Geysers, together with monitored induced seismicity. Low-magnitude seismicity correlates with injection volumes, i.e., the poromechanical response is immediate and mainly local, affecting fractures in the vicinity of wells. Apart from this, there is an increase in the frequency of magnitude 3 earthquakes after 20 years from the start of water injection, and of magnitude 4 earthquakes after 30 years. This shows that the thermomechanical response takes time and that it extends further away than the cooled region, destabilizing distant faults. Similar observations have been shown through numerical simulations of geothermal systems. This is why I was saying that the explained processes at HGF are already known.

Ans: Thank you very much for your comment. During this revision, we found the report on induced seismicity of the National Academy of Sciences (2013) from the following link: <https://www.resolutionmineeis.us/sites/default/files/references/nas-2013.pdf>, and the figure that you mentioned in page 65 of the report (also attached below).

[REDACTED]

From this figure, we see that: (1) the number of EQs with magnitudes of $M \geq 1.5$ (not only low-magnitude seismicity) correlates with injection volumes; (2) the annual number of EQs with magnitudes of $M \geq 3.0$ is very low before 1985 and is around 25 after 1985. However, neither of these phenomena reveal one of our findings that **thermal contraction dominates long-term trends of surface displacement and seismicity growth, while pressure effects drive near-instantaneous changes.**

In addition, our study also has other important findings like: (1) **the pressure effects are relatively stable during the operation history of the HGF whereas the thermal effect increases gradually with time and eventually overwhelms the pressure effects;** (2) **the subsidence anomalies in west and east regions of the HGF during some time periods with net increases of injected fluid are related to the effect of the HGF center siphoning fluid from surrounding regions and ever-increasing effect of thermal contraction, respectively.** Moreover, the framework, developed for resolving the spatiotemporal evolution of the HGF deformation mechanisms through integrating multiple geodetic, geophysical and geological data, is applicable to other complex geothermal systems.